# Strengthening Federated Learning: Surrogate Data-guided Aggregation for Robust Backdoor Defense

## Abstract

Backdoor attacks in federated learning (FL) have garnered significant attention due to their destructive potential. Current advanced backdoor defense strategies typically involve calculating predefined metrics related to local models and modifying the server's aggregation rule accordingly. However, these metrics may exhibit biases due to the inclusion of malicious models in the calculation, leading to defense failures. To address this issue, we propose a novel backdoor defense method in FL named *Su*rrogate *D*ata-guided *A*ggregation (SuDA). SuDA independently evaluates local models using surrogate data, thereby mitigating the influence of malicious models. Specifically, it constructs a surrogate dataset composed of pure noise, which is shared between the server and clients. By leveraging this shared surrogate data, clients train their models using both the shared and local data, while the server reconstructs potential triggers for each local model to identify backdoors, facilitating the filtering of backdoored models before aggregation. To ensure the generalizability of local models across both local and surrogate data, SuDA aligns local data with surrogate data in the representation space, supported by theoretical analysis. Comprehensive experiments demonstrate the substantial superiority of SuDA over previous works.

## 1 Introduction

Federated Learning (FL) (McMahan et al., 2017; Kairouz et al., 2021) is a powerful learning scheme enabling multiple clients to train a global model collaboratively, without leaking their private information. This decentralized nature provides significant advantages over traditional centralized learning, particularly in applications where data privacy is a concern, such as recommendation (Isinkaye et al., 2015; Wu et al., 2021), computer vision (LeCun et al., 1998; Zhu et al., 2020), and healthcare (Xu et al., 2021; Yuan et al., 2020). Although significant progress has been made, FL is still vulnerable to various security threats, such as adversarial attacks. Therefore, how to enable FL to be adversarially robust remains an open question.

This paper focuses on a particular adversarial attack named *backdoor attack* (Chen et al., 2017; Gu et al., 2019; Liao et al., 2018), which is recognized to be very harmful in FL (Bhagoji et al., 2019; Bagdasaryan et al., 2020; Wang et al., 2020a). In general, backdoor attackers manipulate training data on clients, sending local models trained on such tampered data to the server to pollute the global model. Then, after awakening some trigger embeds (i.e., the backdoor) on new inputs, the global model will predict the designated targets given by attackers. For example, an attacker can make the global model predict a specific label (e.g., classify blue trucks as birds) when seeing a particular triggered input (e.g., an image of a blue truck with a particular pattern). The backdoor attack is among the most lethal ways of poisoning (Biggio et al., 2012; Liu et al., 2018) and model stealing (Tramèr et al., 2016; Juuti et al., 2019), posing a great threat to the robustness of real-world FL systems. Hence, it is essential to investigate effective methods for backdoor defense in FL.

Many efforts have been devoted to backdoor defense in FL, where advanced methods mainly focus on the correlation between client models, detecting attacks by analyzing some predefined metrics related to the models themselves, such as Euclidean distance (Blanchard et al., 2017; Pillutla et al., 2022), mean value (Yin et al., 2018), cosine similarity (Fung et al., 2018; Nguyen et al., 2022),

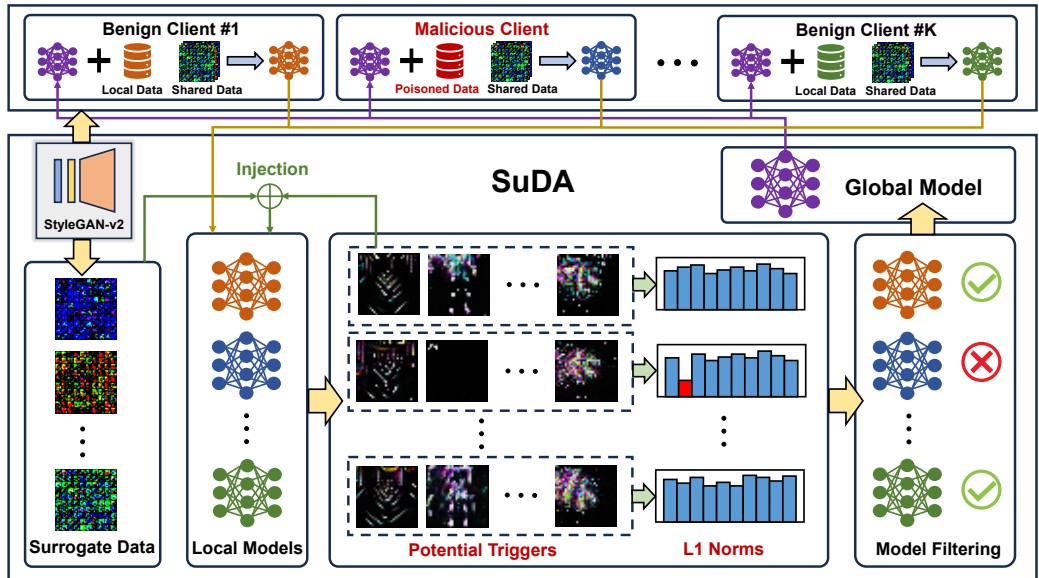

Figure 1: Illustration of SuDA. The SuDA workflow involves introducing a surrogate dataset consisting of pure Gaussian noise and independently evaluating local models. SuDA utilizes this surrogate data to reconstruct potential triggers for each local model, identifying models with abnormally small triggers as malicious. The identified malicious models are then filtered out before aggregation. This metric is independent of the local models, thus mitigating the influence of a large proportion of malicious clients.

and norm bounds (Sun et al., 2019; Panda et al., 2022). However, since malicious models are also involved in the calculation of these metrics, these methods may yield tainted metrics and fail to achieve effective defense. For example, in scenarios where the proportion of malicious clients is significant, these majority-based defense methods may erroneously categorize and exclude the models of the minority benign clients as attackers, thus rendering poor defense performances.

To address the tainted metric issue, a straightforward approach is to design a metric that independently evaluates each local model. To this end, we propose *Su*rrogate *D*ata-guided *A*ggregation (SuDA), a novel backdoor defense method that independently evaluates local models using surrogate data. Overall, SuDA provides a surrogate dataset to the server, thereby allowing the model's performance on the surrogate data to become an effective metric that is independent of local models and mitigating the influence of malicious models. SuDA shares the surrogate dataset across the server and clients, thus clients can train local models with both the local and shared data. To protect privacy, it is often impractical to generate surrogate data that matches the distribution of training data. Therefore, this surrogate dataset is synthesized from pure Gaussian noise data containing no private information from clients (thus without privacy leakages). Leveraging the shared surrogate data, SuDA reconstructs potential triggers for each local model and identifies backdoors based on the size of potential triggers. Consequently, the identified backdoored models can be filtered out before aggregation.

However, simply adding the pure noise data to the training process can potentially impact the performance of local models on the natural data. This is because the original natural data and the noise data have different distributions. To ensure the generalizability of models trained on noise, inspired by the joint distribution alignment (Long et al., 2017), we calibrate the feature distributions of natural data and noise data. Consequently, the noise dataset can represent the training data, enabling the identification of malicious models. We theoretically analyze the relationship between the model's generalization performance and the distribution shift. By completing this task, the noise dataset will not hurt the model training of honest clients while helping the server filter out attackers. The overall framework is depicted in Figure 1. Our comprehensive experiments demonstrate SuDA's stronger defense capabilities compared to previous works.

Our main contributions can be summarized as follows:

- We point out that metrics used for backdoor defense can be tainted by malicious models, leading to the failure of existing approaches.
- We propose a Surrogate Data-guided Aggregation (SuDA) to independently evaluate local models using surrogate data, shedding light on backdoor defense. Specifically, SuDA introduces a surrogate dataset containing pure noise and reconstructs potential triggers with Eq. 4 to identify malicious models.( Section 4)
- Comprehensive experiments on three computer vision datasets demonstrate the effectiveness of SuDA, as shown in Tables 1 and 2. We empirically prove that the proposed metric, which is independent of local models, can help the server accurately filter out malicious models. ( Section 5)

## 2 RELATED WORK

Due to space constraints, we provide an overview of the most relevant works in this section, while a more comprehensive discussion and literature review are available in Appendix B.

Existing FL backdoor defense strategies mainly identify attackers by analyzing specific predefined metrics associated with the models. Blanchard et al. (2017) select model update(s) with the minimum squared distance to the updates of other clients. Coordinate-wise median (Yin et al., 2018) selects the median element coordinate-wise among the model updates of clients. Norm clipping (Sun et al., 2019) clips model updates whose norm exceeds a specific threshold. RFA (Pillutla et al., 2022) replaces the weighted arithmetic mean in FedAvg with a weighted geometric median, which is computed using the *smoothed Weiszfeld's algorithm*. FLAME (Nguyen et al., 2022) eliminates backdoors by injecting noise into the model and employs HDBSCAN clustering and model weight clipping to reduce the required noise. These methods include malicious models in the computation, making it difficult to handle situations where attackers have a large amount of data. Therefore, we propose SuDA, which independently evaluates local models without requiring additional data. Our evaluation includes comparisons to six commonly used defense algorithms and demonstrates the stronger capabilities of SuDA against backdoor attacks.

## 3 PRELIMINARIES

To begin with, we introduce the necessary backgrounds about federated learning ( Section 3.1), backdoor attack ( Section 3.2), and domain adaptation ( Section 3.3).

### 3.1 FEDERATED LEARNING

The federated learning (FL) process is executed by a set of clients in synchronous update rounds, and the server aggregates the local model updates of selected clients in each round to update the global model. Formally, FL aims to minimize a global objective function: $\min_w F(w) := \sum_{k=1}^{K} p_k F_k(w)$, where $K$ is the number of all clients, $p_k \geq 0$ is the weight of $k$-th client, and $F_k$ is the local objective function: $F_k(w) := \mathbb{E}_{(x,y) \sim \mathcal{D}_k(x,y)} \ell(f(x; w), y)$. We denote $\mathcal{D}_k(x, y)$ as the data distribution in the $k$-th client, $\ell(\cdot, \cdot)$ as the loss function such as cross-entropy, and $f$ as the classifier which consist of a feature extractor $\phi$ and a predictor $\rho$, i.e., $f = \rho \circ \phi$.

At each communication round $t$, the server uniformly selects a subset of clients $\mathcal{S}^t$ from the federated system and sends them the current global model $G^t$. The each selected client $k$ performs $E$ epochs local updates to get a new local model $L_k^t$ by training on their private datasets:

$$L_{k,j+1}^t = L_{k,j}^t - \eta_{k,j} \nabla F_k \left( L_{k,j}^t \right), j \in \{0, 1, \cdots, E-1\}, \tag{1}$$

where $\eta_{k,j}$ represents the learning rate, and $L_{k,j}^t$ represents the model after $j$-th updates, i.e., $L_{k,0}^t = G^t$ and $L_{k,E}^t = L_k^t$. Then, all selected clients send the local models back to the server, and the server aggregates these models to produce a global model. Typically, the aggregation is performed using the following sample-based weighting manner (McMahan et al., 2017):

$$G^{t+1} = \sum_{k \in \mathcal{S}^t} \frac{n_k}{\sum_{i \in \mathcal{S}^t} n_i} L_k^t, \tag{2}$$

where $n_k$ is the number of training samples on the $k$-th clients. In federated learning, data distributions typically vary with clients, which is known as the Non-IID federated learning setting, posing a client drift challenge.

## 3.2 BACKDOOR ATTACKS

Backdoor attacks aim to manipulate local models to fit both the main task and the backdoor task simultaneously, inducing the global model to behave normally on untampered data samples while achieving a high attack success rate on backdoored data samples. We consider the *strong attacker* (Bagdasaryan et al., 2020) who can fully control the compromised client, including the private local data and the model training process. When there are multiple attackers, we assume they can collude with each other and share the same target. As discussed in Sun et al. (2019), the participating patterns of attackers can be divided into the *fixed frequency* attack, where the attacker periodically participates in the FL round, and the *random sampling* attack, where the attacker can only perform attacks during the FL rounds in which they are selected. We consider the random sampling case in this paper since this setting is more common in real-life scenarios. The backdoor can also be divided into the *semantic backdoor* (Bagdasaryan et al., 2020; Wang et al., 2020a), which denotes samples that share the same semantic property, and the *trigger-based backdoor* (Xie et al., 2020), which denotes samples that contain a specific "trigger". Here we consider the trigger-based backdoor attacks following previous works (Xie et al., 2020; Zhang et al., 2022). Furthermore, we form the backdoor task by conducting model replacement attacks introduced in Bagdasaryan et al. (2020).

## 3.3 DOMAIN ADAPTATION

The core challenge in domain adaptation is how to address the impact of the inconsistency between the distribution of training data and testing data (Pan & Yang, 2010), referred to as the source domain and the target domain. Distribution shift can be classified according to the components that cause the shift into covariate shift (Pan et al., 2010; Ben-David et al., 2010), conditional shift (Zhang et al., 2013; Gong et al., 2016), and dataset shift (Quinonero-Candela et al., 2008; Long et al., 2013; Zhang et al., 2020), corresponding shifts for $\mathcal{D}(x)$, $\mathcal{D}(x|y)$ and $\mathcal{D}(x,y)$ respectively.

A commonly used and effective method for reducing the impact of distribution shift is to use a feature extractor $\phi$ to extract similar feature distributions from the source distribution $\mathcal{D}_S$ and the target distribution $\mathcal{D}_T$ (Ganin et al., 2016; Zhao et al., 2019; Long et al., 2017). Specifically, the feature extractor $\phi$ minimizes the distribution discrepancy for three types of distribution shift with the measurement $d$ respectively (Ganin et al., 2016; Gong et al., 2016): the marginal distribution discrepancy $d(\mathcal{D}_S(\phi(x)), \mathcal{D}_T(\phi(x)))$, the conditional distribution discrepancy $d(\mathcal{D}_S(\phi(x)|y), \mathcal{D}_T(\phi(x)|y))$ and the joint distribution discrepancy $d(\mathcal{D}_S(\phi(x), y), \mathcal{D}_T(\phi(x), y))$. In this paper, we need to consider the most challenging dataset shift and minimize the joint distribution discrepancy. We regard the surrogate dataset as the source domain and the original local dataset as the target domain.

## 4 SURROGATE DATA-GUIDED AGGREGATION STRATEGY

This section proposes a novel Surrogate Data-guided Aggregation (SuDA) approach to defend against backdoor attacks in FL by introducing a special surrogate dataset containing pure noise to assist the server in identifying and filtering out malicious models.

### 4.1 OVERVIEW OF SuDA

In the proposed SuDA framework, the server crafts a surrogate dataset that consists of pure Gaussian noise generated by an untrained Style-GAN (Karras et al., 2019). Then all clients receive the surrogate dataset and train local models with the objective Eq. 7. Thereby, the server can utilize the shared surrogate data to reconstruct potential triggers for each local model independently with the objective Eq. 4 and identify malicious models based on the size of potential triggers. Different from previous methods, this metric is independent of local models and impervious to variations in the attacker's ratio. Therefore, SuDA demonstrates better defensive performance when the proportion of malicious clients is significant. The overall procedure of SuDA coupled with FedAvg is illustrated in Appendix D. For the client, SuDA only modifies its training data and objective function, whereas

for the server, SuDA only introduces additional model filtering steps based on the surrogate dataset. Therefore, SuDA can be combined with most federated learning algorithms, including FedAvg, Fed-Prox (Li et al., 2020) and FedNova (Wang et al., 2020c). Note that malicious attackers may reject to follow the proposed protocol. Therefore, we consider several different adaptive attackers and empirically prove that SuDA can effectively defend against malicious clients under adaptive attack scenarios in Section 5.

## 4.2 RECONSTRUCT POTENTIAL TRIGGER

The failure of existing methods can be mainly attributed to the tainted metrics used for filtering out malicious models. Specifically, existing methods for defending against backdoor attacks in FL focus on studying the attributes of the received client models themselves, such as taking the mean or median of the model parameters (Yin et al., 2018), filtering out outliers based on squared-distance of updates (Blanchard et al., 2017), and clipping updates with excessive norms (Sun et al., 2019). Consequently, these methods encounter a dilemma when the proportion of malicious clients is significant: these metrics will become unreliable and render poor defense performances.

A more direct approach to backdoor defense is to independently evaluate each local model. Drawing inspiration from the insights presented in the centralized setting (Wang et al., 2019), our objective is to reconstruct potential triggers from the model. By perturbing the input pixels, we can manipulate the model's output for a given input sample. Specifically, consider a model that has been compromised with a trigger targeting a specific label $Y_t$. Note that the trigger should be reasonably small, otherwise it will be easily detected. For any arbitrary inputs, regardless of their true label $Y_i$, we can observe that the minimum perturbation required to classify all inputs as the target label is significantly smaller than the perturbation needed to transform the inputs to any non-target label.

**Observation 4.1.** *If there is a trigger with a target label $Y_t$, then the minimum perturbation required to classify all inputs as the target label should be significantly smaller than the perturbation needed to transform the inputs to any non-target label: $P_{\forall \to t} << \min_{i, i \neq t} P_{\forall \to i}$.*

Based on the observation above, we can treat each label as a potential target label and calculate the minimum potential trigger required to misclassify samples of other labels into this target label. Specifically, we represent the process of injecting a trigger into the input $x$ as follows:

$$x_{poison} = M \cdot \Delta + (1 - M) \cdot x, \tag{3}$$

where $M$ is a trigger mask with values ranging from 0 to 1, $\Delta$ is a trigger pattern that has the same dimension as the input image. We use the $L1$ norm of the mask $M$ to measure the size of the trigger. Our goal is to find a trigger that can misclassify clean samples to the target label while being as small as possible. Consequently, we can reconstruct the potential trigger by optimizing the following objective:

$$\min_{M, \Delta} \ell \left( Y_t, f(x_{poison}) \right) + \gamma \cdot |M|, \tag{4}$$

where $\gamma$ is a parameter that balances the misclassification success rate and the size of the trigger. In the process of optimization, we dynamically adjust the parameter $\gamma$ to gradually achieve a more concise trigger.

Using the optimization objective, we reconstruct the potential trigger for each label. For all potential triggers obtained, we perform *Median Absolute Deviation* outlier detection on their $L1$ norms. If there is a significantly small outlier, we identify the corresponding label as the attacker's target label and recognize the current model as a malicious model; otherwise, we recognize the current model as a benign model.

In the above method, however, the server needs data to reconstruct potential triggers, which is a challenge in the federated setting. In order to protect privacy, the client cannot directly share local training data with the server. Therefore, we propose to construct a surrogate dataset that contains no private information. The server sends the surrogate dataset to all clients and requires them to train local models with both the original local data and the shared surrogate data:

$$F_k^{cls} := \mathbb{E}_{(x,y) \sim \mathcal{D}_k} \ell(f(x), y) + \mathbb{E}_{(x,y) \sim \mathcal{D}_n} \ell(f(x), y), \tag{5}$$

where $\mathcal{D}_n$ is the distribution of the surrogate dataset. Since the surrogate data cannot contain any private information, we propose that the surrogate dataset can be generated using pure Gaussian noise, such as random noise derived from a randomly initialized StyleGAN (Karras et al., 2019). The noise data shares the same range of labels as the real data, with each label denoting a different style of noise. With the surrogate data, the server uniformly selects $b$ samples from the surrogate dataset each round to reconstruct potential triggers. This objective function is formulated to ensure that the local client model performs well on both the real and surrogate data, serving as a crucial reference for the server to detect potential attackers. Given that noise has corresponding labels, we can preliminarily filter out models with excessively low prediction accuracy on the noise data before reconstructing potential triggers. This measure aims to prevent attackers from uploading excessively modified models.

## 4.3 FEATURE DISTRIBUTION ALIGNMENT

Intuitively, simply adding a surrogate dataset that has a completely different distribution from the original local data will harm the model's generalization performance (Frénay & Verleysen, 2013; Polyzotis et al., 2017). Therefore, inspired by the previous work (Long et al., 2017), which investigates the joint distribution discrepancy, we further introduce feature distribution alignment to enable the models trained on surrogate data to perform well on the natural distribution.

To effectively represent real data using surrogate data, it is also crucial to align the distribution of real features with that of surrogate ones. To facilitate the transfer of knowledge from the surrogate dataset to the real dataset, we draw inspiration from domain adaptation techniques. Specifically, we consider the surrogate dataset as the source domain and the real dataset as the target domain and perform domain adaptation to mitigate the generalization risk of the real distribution. By leveraging the fundamental principles of domain adaptation, we align these two distributions in the feature space and ensure the good performance of the model trained with surrogate data on real data.

Following the previous works on addressing dataset shift (Long et al., 2013; 2017; Lei et al., 2021), we minimize the joint distribution discrepancy between real features and surrogate features. Note that the surrogate dataset is arbitrarily constructed, this allows us to generate appropriate noise data with the same label distribution as the real data. Consequently, we only need to minimize the conditional distribution discrepancy rather than the joint distribution discrepancy. In particular, We propose the objective for the feature distribution alignment:

$$F_k^{da} := \mathbb{E}_y d(\mathcal{D}_k(\phi(x)|y), \mathcal{D}_n(\phi(x)|y)), \tag{6}$$

where $\phi$ is the feature extractor that composes the classifier $f = \rho \circ \phi$, $\mathcal{D}_k(\phi(x)|y)$ and $\mathcal{D}_n(\phi(x)|y)$ represent the conditional feature distributions obtained by the feature extractor $\phi$ on the real dataset and the surrogate dataset, respectively. This objective encourages the feature extractor to learn the same conditional feature distribution from two different data distributions.

Thereafter, we propose the overall objective of clients during local training in the SuDA framework:

$$F_k^{SuDA} = F_k^{cls} + \lambda F_k^{da}, \tag{7}$$

where $\lambda$ is a hyperparameter that governs the trade-off between classification accuracy on training data and the degree of alignment in feature distribution. SuDA enables the model to accurately classify real data and noise data, and simultaneously encourages the model to generate similar features from real data and noise data with the same label, thus achieving good generalization performance on both distributions. Empirical observations in Figure 8 demonstrate this effect.

To theoretically prove the effectiveness of the proposed feature distribution alignment, we analyze the relationship between the model's generalization performance and the distribution shift. Based on the existing theoretical conclusions, the generalization performance is related to the margin between samples and the decision boundary. Therefore, we introduce the definition of *statistical robustness* between two distributions before stating the theorem, serving as a metric for quantifying the degree of generalization performance.

**Definition 4.2** (Statistical Robustness)**.** *We define statistical robustness for a classifier $f$ on a distribution $\mathcal{D}$ according to a distance metric $d$: $SR_d(f, \mathcal{D}) = \mathbb{E}_{(x,y)\sim\mathcal{D}} \inf_{f(x')\neq y} d(x', x)$, where classifier $f : \mathcal{X} \to \mathcal{Y}$ predicts class label of an input sample.*

The defined statistical robustness refers to the expected distance from each sample to the closest adversarial example. Hence, for the model $f$ learned from the source distribution $\mathcal{D}_n(x, y)$, we can quantify the generalization performance on the target distribution $\mathcal{D}(x, y)$. To achieve good generalization performance, we aim to provide a lower bound on the transferred statistical robustness, i.e., $\mathbb{E}_{\substack{\mathcal{S} \sim \mathcal{D}_n \\ f \leftarrow Sub(\mathcal{S})}} SR_d(f, \mathcal{D})$, where $f \leftarrow Sub(\mathcal{S})$ means the model $f$ is trained on the training set $\mathcal{S}$ using SuDA. To this end, we have the following theorem.

**Theorem 4.3.** *Let $f$ be a neural network, $\mathcal{D}(x, y)$ and $\mathcal{D}_n(x, y)$ are two separable distributions with identical label distributions, corresponding to the distributions of real data and noise data, respectively. Then, training the model with the proposed objective for the feature distribution alignment, i.e., Eq. 7 elicits the bounded statistical robustness.*

We provide the proof of Theorem 4.3 in Appendix A.1. Theorem 4.3 shows that the model trained with the proposed objective can learn to provable generalization performance, which is consistent with the previous work (Long et al., 2013; 2017) that aligning the joint distribution between the source and target domains.

## 5 EXPERIMENTS

The goal of our empirical study is to demonstrate the improved defense capability of SuDA over the state-of-the-art FL defense methods. We conduct our experiments on image classification tasks over three datasets: CIFAR-10(Krizhevsky et al., 2009), FMNIST(Xiao et al., 2017) and SVHN(Netzer et al., 2011). We simulate FL for $R$ rounds among $K$ clients, of which $m$ are corrupted by attackers. In each round, the server uniformly selects $C \cdot K$ clients for some $C \leq 1$ and sends the global model to each selected client. The selected clients then perform local training on the received model for $E$ epochs and send the updates back to the server. The goal of attackers is to make the aggregated model misclassify samples poisoned by triggers into the target class. For the aggregated model on the server, we measure three performance metrics: total accuracy, attack success rate and main accuracy. Total accuracy is computed on the entire test dataset, while attack success rate measures the proportion of test samples with triggers classified as the target label by the model, and main accuracy is computed on clean test samples. Our experimental results show that SuDA significantly outperforms baseline methods in defending against backdoor attacks.

### 5.1 EXPERIMENT SETUP

**Datasets and Models.** To evaluate the effectiveness of SuDA, we conduct experiments on three computer vision datasets including CIFAR-10, FMNIST and SVHN in the FedML framework(He et al., 2020). We use ResNet-18(He et al., 2016) as the shared global model in FL for all three datasets. We utilize the partition method Latent Dirichlet Sampling(Hsu et al., 2019) to partition datasets, generating a local dataset for each client, and using the parameter $\alpha$ to control the degree of Non-IID. We set $\alpha = 1$ to simulate the Non-IID setting by default and conduct experiments under the IID setting in Appendix E.3.

**Surrogate Datasets.** At the beginning of the training phase, an un-pretrained StyleGAN-v2 (Karras et al., 2020) is utilized to generate a surrogate dataset without using any training data. The server samples from various Gaussian distributions, each with the same mean but different standard deviations, to generate noise images with diverse latent styles. Each style corresponds to a distinct class. Then the generated noise images are distributed to all clients and used together with the original datasets for local training. The size of the surrogate dataset in our experiments is 2000. We show the surrogate dataset in Appendix E.4.

**Random sampling attack.** The attack model considered in our work is the random sampling attack as discussed in Sun et al. (2019), where the attackers have complete control over a fraction of clients. In each FL round, the server randomly selects a subset of clients to participate in the training process. The attackers are only able to affect the training of the global model during the rounds in which they are selected. The number of selected attackers in each round follows a hypergeometric distribution.

**Backdoor tasks.** The backdoor task aims to make the global model misclassify backdoored samples into the target class. Since the server randomly selects clients in each round, multiple attackers may be chosen during a single round. We assume that attackers can collude and share the same target,

Table 1: ACC, ASR and MA of defense algorithms on CIFAR-10, FMNIST and SVHN when defending against varying attackers. The poison ratio is 5%.

| Atk Num | Defense | CIFAR-10 | | | FMNIST | | | SVHN | | |
|---|---|---|---|---|---|---|---|---|---|---|
| | | ACC(%) | ASR(%) | MA(%) | ACC(%) | ASR(%) | MA(%) | ACC(%) | ASR(%) | MA(%) |
| 0 | FedAvg | 84.72±0.92 | 3.26±0.85 | 84.87±0.84 | 91.51±0.09 | 1.77±0.64 | 91.64±0.10 | 89.20±0.02 | 1.12±0.01 | 89.28±0.02 |
| | RFA | 84.92±0.95 | 1.44±0.26 | 84.99±0.94 | 91.69±0.02 | 1.88±0.28 | 91.85±0.03 | 89.22±0.02 | **0.98±0.04** | 89.27±0.02 |
| | Krum | 50.24±1.25 | 3.88±0.91 | 49.85±1.13 | 86.69±0.62 | 3.57±0.77 | 86.79±0.57 | 79.65±0.18 | 1.97±0.74 | 79.68±0.24 |
| | Coomed | 83.92±0.84 | **1.16±0.33** | 83.94±0.86 | **91.79±0.02** | 1.95±0.16 | **91.88±0.02** | 88.99±0.06 | 1.19±0.02 | 89.07±0.05 |
| | Normclip | 85.07±0.62 | 1.38±0.16 | 85.18±0.60 | 91.54±0.03 | 1.68±0.14 | 91.66±0.04 | 87.56±0.01 | 1.46±0.05 | 87.67±0.01 |
| | FLAME | 83.39±0.94 | 1.18±0.19 | 83.44±1.07 | 91.04±0.01 | 1.86±0.09 | 91.20±0.01 | 87.11±0.32 | 1.68±0.23 | 87.30±0.29 |
| | FLTrust | 68.30±0.91 | 2.91±0.78 | 68.28±0.25 | 86.58±0.93 | 1.33±0.62 | 86.67±0.47 | 87.90±0.12 | 1.35±0.06 | 88.02±0.12 |
| | SuDA(ours) | **85.73±0.31** | 1.75±0.40 | **85.78±0.33** | 90.09±0.38 | **1.29±0.74** | 90.17±0.44 | **90.78±0.70** | 1.20±0.39 | **90.88±0.31** |
| 4 | FedAvg | 76.79±0.81 | 87.63±0.77 | 83.40±1.00 | 83.71±0.03 | 99.78±0.01 | **92.09±0.04** | 81.41±0.42 | 67.14±0.81 | 86.66±0.13 |
| | RFA | 79.38±0.23 | 76.31±0.58 | 85.20±0.39 | 87.41±0.91 | 22.74±0.50 | 89.20±0.86 | 88.99±0.04 | 5.36±0.48 | 89.36±0.04 |
| | Krum | 49.58±0.18 | 5.50±1.68 | 49.41±0.67 | 84.12±0.11 | 5.30±0.89 | 84.23±0.98 | 83.51±0.19 | 1.26±0.29 | 83.55±0.22 |
| | Coomed | 76.75±0.13 | 58.44±1.48 | 80.90±0.39 | 88.93±0.24 | 5.83±0.44 | 89.36±0.35 | 89.04±0.16 | 3.09±0.20 | 89.25±0.09 |
| | Normclip | 79.41±0.20 | 78.17±0.64 | 85.41±0.27 | 88.94±0.36 | 8.55±0.31 | 89.58±0.24 | 87.58±0.03 | 2.19±0.04 | 87.7±0.03 |
| | FLAME | 77.19±1.08 | 82.65±0.79 | 83.36±0.41 | 82.70±0.13 | 96.68±1.00 | 90.71±0.09 | 84.13±0.82 | 17.29±1.01 | 85.31±0.79 |
| | FLTrust | 65.54±0.49 | 38.99±0.33 | 67.57±0.72 | 88.27±1.01 | 4.97±0.17 | 88.82±1.33 | 86.47±0.06 | 11.34±0.97 | 87.23±0.12 |
| | SuDA(ours) | **84.65±0.12** | **2.38±0.56** | 84.77±0.43 | **90.80±0.17** | **1.20±0.99** | 90.92±0.16 | **90.74±0.60** | **1.22±0.67** | **90.76±0.38** |
| 8 | FedAvg | 77.86±0.18 | 88.9±0.22 | 84.67±0.61 | 83.27±0.07 | 99.67±0.06 | **91.61±0.08** | 80.79±0.98 | 66.21±0.60 | 85.9±0.85 |
| | RFA | 78.69±0.82 | 83.42±1.01 | 85.07±1.11 | 84.43±0.55 | 30.17±0.89 | 86.80±0.05 | 87.59±0.02 | 4.65±0.26 | 87.88±0.03 |
| | Krum | 50.91±1.03 | 6.44±1.12 | 50.79±0.90 | 81.87±0.32 | 3.60±0.75 | 82.00±0.38 | 80.55±0.34 | 1.36±0.03 | 80.55±0.38 |
| | Coomed | 77.27±0.72 | 81.88±0.93 | 83.42±0.41 | 84.68±0.71 | 10.17±0.13 | 85.49±0.09 | 86.83±0.48 | 7.05±0.59 | 87.25±0.41 |
| | Normclip | 78.85±0.51 | 83.33±1.27 | 85.26±0.71 | 86.90±0.48 | 7.78±0.51 | 87.50±0.55 | 87.34±0.03 | 5.16±0.20 | 87.69±0.02 |
| | FLAME | 76.87±0.31 | 86.71±1.02 | 83.38±0.46 | 82.56±0.04 | 99.16±0.04 | 90.78±0.06 | 80.05±0.23 | 82.86±0.73 | 86.63±0.22 |
| | FLTrust | 68.59±1.15 | 66.75±0.96 | 72.88±0.94 | 74.14±1.60 | 96.33±1.20 | 81.25±0.55 | 81.82±0.20 | 66.16±0.15 | 86.95±0.22 |
| | SuDA(ours) | **84.56±0.65** | 3.22±0.67 | 84.73±0.69 | **90.75±0.10** | **1.31±0.24** | 90.79±0.14 | **90.70±0.40** | **1.16±0.38** | **90.73±0.34** |
| 12 | FedAvg | 77.80±1.07 | 89.64±0.64 | 84.67±0.77 | 83.34±0.17 | 99.64±0.02 | **91.68±0.20** | 81.58±0.14 | 75.29±0.42 | 87.61±0.11 |
| | RFA | 77.90±0.69 | 87.76±0.03 | 84.61±0.84 | 81.88±0.59 | 44.36±0.85 | 85.32±0.93 | 86.13±0.23 | 8.12±0.64 | 86.64±0.16 |
| | Krum | 48.03±0.67 | 8.47±0.96 | 47.9±0.50 | 80.02±0.15 | 10.89±0.49 | 80.51±0.80 | 80.74±0.14 | 2.60±0.05 | 80.80±0.33 |
| | Coomed | 77.12±0.79 | 84.62±0.79 | 83.49±1.03 | 83.31±0.06 | 26.27±0.10 | 83.80±0.38 | 86.47±0.43 | 13.57±0.10 | 87.39±0.18 |
| | Normclip | 78.68±0.42 | 87.91±0.38 | 85.47±0.55 | 85.24±0.41 | 15.08±0.46 | 86.40±0.65 | 86.31±0.31 | 13.39±0.39 | 87.18±0.18 |
| | FLAME | 76.92±0.23 | 88.75±0.83 | 83.60±0.23 | 82.45±0.27 | 99.62±0.01 | 90.69±0.33 | 79.48±0.34 | 86.09±1.18 | 86.32±0.36 |
| | FLTrust | 58.65±0.39 | 78.02±0.69 | 62.85±0.75 | 79.04±0.68 | 98.96±0.44 | 86.89±0.52 | 80.90±0.04 | 78.43±0.18 | 87.13±0.05 |
| | SuDA(ours) | **83.89±0.43** | **3.71±0.57** | 84.11±0.59 | **89.75±0.15** | **1.44±0.80** | 89.88±0.33 | **89.79±0.36** | **1.63±0.11** | **89.83±0.78** |

Table 2: Performance of SuDA under changing attack $\lambda$ on CIFAR-10.

| Atk Num | Metrics | Atk $\lambda$ | | | | | | | |
|---|---|---|---|---|---|---|---|---|---|
| | | 0 | 0.1 | 0.5 | 1 | 2 | 5 | 10 | 100 |
| 4 | ACC(%) | 85.21±0.78 | **85.26±0.97** | 85.09±0.81 | 85.20±0.55 | 84.96±0.71 | 85.21±0.22 | 85.15±0.98 | 79.75±0.50 |
| | ASR(%) | 2.58±0.44 | 2.21±0.79 | 2.37±0.47 | 2.30±0.98 | **2.14±0.91** | 2.60±0.94 | 2.41±0.72 | 2.30±0.34 |
| | MA(%) | 85.29±0.60 | 85.36±0.81 | 85.14±0.70 | 85.31±0.19 | 85.14±0.70 | 85.36±0.36 | **85.40±0.51** | 79.79±0.65 |
| 8 | ACC(%) | 84.35±0.87 | 85.05±0.68 | 84.91±0.37 | **85.25±0.93** | 85.00±0.96 | 85.05±0.74 | 84.72±0.28 | 72.77±0.23 |
| | ASR(%) | **2.01±0.60** | 2.30±0.35 | 2.20±0.72 | 2.49±0.62 | 2.26±0.42 | 2.36±0.10 | 2.17±0.76 | 2.01±0.38 |
| | MA(%) | 84.49±0.94 | 85.21±0.44 | 85.01±0.47 | **85.42±0.85** | 85.15±0.84 | 85.18±0.62 | 84.90±0.98 | 72.62±0.62 |

i.e. all attackers aim to make the global model misclassify backdoored samples into the same target class. For the CIFAR-10 and FMNIST datasets, attackers aim to misclassify into class '2', and for the SVHN dataset, attackers aim to misclassify into class '5'. In each round, attackers implant the trigger into partial samples of each class based on the poison ratio, re-label them with the target class, and then train the local model on the backdoored dataset. The trigger utilized is a white square measuring $4 \times 4$, implanted in the upper-left corner of the poisoned sample. When employing SuDA for defense, the noise dataset will be combined with the backdoored original local dataset for training. Attackers further perform model replacement attacks(Bagdasaryan et al., 2020) to generate malicious local models and send them to the server.

**Defense techniques.** We conduct FedAvg(McMahan et al., 2017) as the baseline FL aggregation algorithm. The results using FedProx are reported in Appendix E.2. To demonstrate the effectiveness of SuDA in defending against backdoor attacks, we consider six commonly used defense techniques: (i) Krum (Blanchard et al., 2017); (ii) Coordinate-wise median(Coomed) (Yin et al., 2018); (iii) Norm clipping(Normclip) (Sun et al., 2019); (iv) RFA (Pillutla et al., 2022); (v) FLAME (Nguyen et al., 2022) and (vi) FLTrust (Cao et al., 2020). The detailed hyper-parameters of these algorithms are reported in Appendix C.

## 5.2 EXPERIMENTAL RESULTS

To compare the performance of different defense algorithms, we use three metrics: the average total accuracy (ACC), the average attack success rate (ASR), and the average accuracy of main tasks (MA) in the 5 rounds before the model converges. We conduct FL with a maximum of 200 rounds

Table 3: Results of reconstructing potential triggers without the shared surrogate data.

| Atk Num | Defense | CIFAR-10 | | | FMNIST | | | SVHN | | |
|---|---|---|---|---|---|---|---|---|---|---|
| | | ACC(%) | ASR(%) | MA(%) | ACC(%) | ASR(%) | MA(%) | ACC(%) | ASR(%) | MA(%) |
| 4 | Noise Recon. | 75.05±0.34 | 74.22±0.04 | 80.33±0.22 | 85.52±0.18 | 19.33±0.12 | 88.44±0.49 | 88.26±0.77 | 4.01±0.62 | 89.40±0.34 |
| | SuDA(ours) | **84.65±0.12** | **2.38±0.56** | **84.77±0.43** | **90.80±0.17** | **1.20±0.99** | **90.92±0.16** | **90.74±0.60** | **1.22±0.67** | **90.76±0.38** |
| 8 | Noise Recon. | 76.29±0.90 | 75.49±0.68 | 81.21±0.38 | 82.06±0.28 | 24.40±0.82 | 85.85±0.76 | 86.70±0.82 | 4.91±0.48 | 86.98±0.22 |
| | SuDA(ours) | **84.56±0.65** | **3.22±0.67** | **84.73±0.69** | **90.75±0.10** | **1.31±0.24** | **90.79±0.14** | **90.70±0.40** | **1.16±0.38** | **90.73±0.34** |

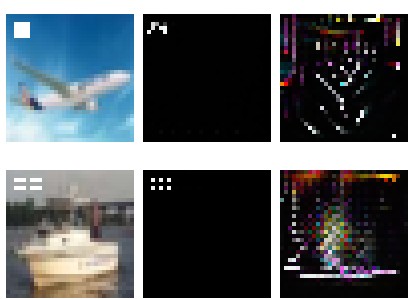

Figure 2: Potential triggers for the target label (second column) and non-target labels (third column) reconstructed from two different types of poisoned data.

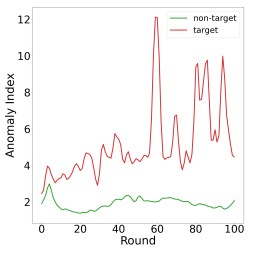

(a) $L1$ Norm      (b) Anomaly Index

Figure 3: The changes in the $L1$ norm and anomaly index of potential triggers corresponding to target and non-target labels during the training process.

using the adopted defense algorithm on CIFAR-10, FMNIST and SVHN. There are 50 clients in total, the number of backdoor attackers can range from 0 to 12, and the poison ratio can range from 1% to 20%, depending on different settings. In each round, the server randomly selects 20 clients to participate in training and sends them the global model. The selected clients then perform local training for 1 epoch on the received model and send the locally trained model to the server.

**Different Numbers of Attackers.** As shown in Table 1, SuDA outperforms other baselines in almost all scenarios when defending against varying numbers of attackers across the three datasets. It can be seen that SuDA can improve model performance even without attackers. We conjecture that this is mainly due to several factors: 1) Adding the surrogate dataset reduces data heterogeneity between clients and mitigates client drift; 2) Aligning real feature distribution with the shared noise feature distribution further mitigates client drift; 3) After adding the surrogate dataset, the clients' training data contains more classes, which can alleviate the negative impact caused by the imbalance of sample quantities among different classes.

When there are attackers in FL, SuDA's performance is also superior to other baselines. SuDA achieves a significantly lower ASR than other baselines while maintaining high model accuracy. In particular, when the number of attackers is 12, SuDA reduces the ASR by up to 9.45% compared to the second-ranked method, showing the effectiveness of SuDA in defending against a large number of malicious attackers. We notice that the MA of SuDA is sometimes slightly lower than the best result. We speculate that this is mainly because SuDA filters out malicious models before aggregation, reducing the number of models aggregated. Consequently, the aggregated model becomes more difficult to converge, especially in Non-IID settings.

**Reconstruct Potential Triggers.** SuDA reconstructs potential triggers by leveraging the shared surrogate data. Figure 3a shows the $L1$ norm of potential triggers, and the red dots represent potential triggers for the target label. It can be seen that the $L1$ norm of the potential triggers corresponding to the target label is significantly smaller than that of the triggers corresponding to non-target labels. Accordingly, we perform *Median Absolute Deviation* outlier detection on the $L1$ norms and filter out triggers with an excessive anomaly index, which is defined as the absolute deviation of $L1$ norms divided by MAD. The results in Figure 3b empirically demonstrate that we can distinguish between target and non-target label triggers using anomaly index. We show the reconstructed potential triggers for the target label and non-target labels in the second and third columns of Figure 2, respectively.

**Impact of Surrogate Data.** To further illustrate the impact of the introduced surrogate dataset, we conducted experiments without using the surrogate data, instead utilizing completely random noise, which does not participate in model training, to reconstruct potential triggers. The experimental results are shown in Table 3. As demonstrated in the table, the performance of using noise data that does not participate in training is worse than that of using the shared surrogate data. This further shows that the alignment operation employed by SuDA enables the surrogate data to represent the real data better, thereby reconstructing the potential triggers more accurately and achieving better defense performance.

**Adaptive Attack Scenario.** Note that malicious attackers may reject following the proposed framework and attempt to circumvent SuDA. To further illustrate SuDA's defensive capabilities, we assume that the attackers have knowledge about SuDA and adopt more specialized attack methods. Specifically, we consider the following adaptive attack scenario: the attacker changes the alignment parameter $\lambda$ used to a different one not used by the benign clients. The experimental results are shown in Table 2. It can be seen that SuDA still performs relatively well against this adaptive attack, which demonstrates SuDA's robust defense capabilities against malicious attackers employing various attack methods. We conduct more experiments under adaptive attack scenarios in Appendix E.7.

**Extra Time Overhead.** In our proposed SuDA, reconstructing potential triggers inevitably introduces additional computational overhead. To minimize the extra time overhead while maintaining defensive effects, we record the optimized trigger obtained in each round as the initial value for the next round of optimization, eliminating the need to estimate from scratch. Additionally, we use early-stop in trigger optimization to reduce time consumption. When rounds surpass 60, early-stop significantly reduces the extra time overhead. To further reduce time overhead, we also investigate a more efficient method, SuDA-Efficient. SuDA-Efficient achieves a lower time overhead at the expense of a slight reduction in defense capability. These strategies make the extra time overhead of our method acceptable. To demonstrate the trade-off between SuDA's defensive capability and time overhead, we conduct experiments to compare SuDA and baseline methods on CIFAR-10. The experimental results are shown in Table 14, with more detailed information reports in Appendix E.9.

**More Experimental Results.** In order to further investigate the effectiveness, applicability, and scalability of SuDA, we conduct more ablation experiments. We investigate the defensive performance of SuDA under different poison ratios and the impact of various surrogate data generation methods on the performance of SuDA. We also conduct ablations on the effect of the size of the surrogate dataset and the sensitivity of sample number $b$ of the surrogate dataset. We report these experimental results and more ablation experiments in Appendix E.

## 5.3 LIMITATIONS

Although our method achieves significant improvement in the experiment, it also introduces additional communication overhead. To mitigate this overhead, we hope to minimize the noise dataset as much as possible. However, the size of the noise dataset may also affect the performance of the client model and the server's ability to detect backdoor attacks accurately. Therefore, future research should investigate the optimal size of the noise dataset that strikes a balance between communication overhead and model performance.

## 6 CONCLUSION

In this paper, we introduce a generated noise dataset that does not contain real data information into the defense against backdoor attacks in FL. These surrogate noise data provide a more direct and accurate metric for the server to detect malicious models. Through the conditional feature distribution alignment on the noise dataset, our proposed SuDA can effectively filter out malicious models on the server with the assistance of noise data, without affecting the generalization performance of the local model trained by benign clients. Our empirical results demonstrate that SuDA can effectively defend against backdoor attacks and improve the performance of aggregated models, especially when the proportion of malicious clients is significant, providing new insights for defending against attacks in FL. We hope that our work will inspire further research in developing effective defense mechanisms for FL and contribute to the broader goal of securing machine learning systems.

## ETHIC STATEMENT

This paper does not raise any ethical concerns. This study does not involve any human subjects, practices to data set releases, potentially harmful insights, methodologies and applications, potential conflicts of interest and sponsorship, discrimination/bias/fairness concerns, privacy and security issues, legal compliance, and research integrity issues.

## REPRODUCIBILITY STATEMENT

To make all experiments reproducible, we have listed all detailed hyper-parameters of each FL algorithm. Due to privacy concerns, we will upload the anonymous link of source codes and instructions during the discussion phase to make it only visible to reviewers.

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

# A  PROOF

## A.1  PROOF FOR THEOREM 4.3

*Proof.* We decompose the statistical robustness $SR_d(f, \mathcal{D}(x, y))$ to three quantities as follows:

$$SR_d(f, \mathcal{D}) = (SR_d(f, \mathcal{D}) - SR_d(f, \mathcal{D}_n)) + (SR_d(f, \mathcal{D}_n) - SR_d(f, \tilde{\mathcal{D}}_n)) + SR_d(f, \tilde{\mathcal{D}}_n), \quad (8)$$

where $\tilde{\mathcal{D}}_n$ denotes the empirical distribution for the training set sampled from the noise data distribution $\mathcal{D}_n$. Then based on the linearity of expectation and triangle inequality, we can bound the transferred statistical robustness as follows:

$$\mathbb{E}_{f \leftarrow \mathcal{D}} SR_d(f, \mathcal{D}) \geq \mathbb{E}_{f \leftarrow \mathcal{D}} SR_d(f, \tilde{\mathcal{D}}_n) - |\mathbb{E}_{f \leftarrow \mathcal{D}}[SR_d(f, \mathcal{D}_n) - SR_d(f, \tilde{\mathcal{D}}_n)]| \\ - |\mathbb{E}_{f \leftarrow \mathcal{D}}[SR_d(f, \mathcal{D}) - SR_d(f, \mathcal{D}_n)]|, \quad (9)$$

where $\mathbb{E}_{f \leftarrow \mathcal{D}}$ denotes $\mathbb{E}_{\substack{\mathcal{S} \sim \mathcal{D} \\ f \leftarrow Nog(\mathcal{S})}}$ for brevity. The three terms above represent the empirical robustness, the generalization penalty and the distribution shift penalty, respectively. Since our goal is to bound the transferred statistical robustness, we need to bound both the generalization penalty and the distribution shift penalty. There are already multiple works (Diochnos et al., 2018; Schmidt et al., 2018; Montasser et al., 2019) have studied the bound of the generalization penalty. In order to bound the distribution shift penalty, we introduce the following lemma:

**Lemma A.1.** *Let $\mathcal{D}$ and $\mathcal{D}_n$ be two distributions with identical label distributions, $d(\cdot, \cdot)$ be the Wasserstein distance of two distributions. Then for any classifier $f$, we have:*

$$|SR_d(f, \mathcal{D}) - SR_d(f, \mathcal{D}_n)| \leq \mathbb{E}_y d(\mathcal{D}|y, \mathcal{D}_n|y). \quad (10)$$

We prove Lemma A.1 in the follow-up section, i.e., Appendix A.2. With Eq. 9 and Lemma A.1, we can further bound the transferred statistical robustness as follows:

$$\mathbb{E}_{f \leftarrow \mathcal{D}} SR_d(f, \mathcal{D}) \geq \mathbb{E}_{f \leftarrow \mathcal{D}} SR_d(f, \tilde{\mathcal{D}}_n) - |\mathbb{E}_{f \leftarrow \mathcal{D}}[SR_d(f, \mathcal{D}_n) - SR_d(f, \tilde{\mathcal{D}}_n)]| \\ - \mathbb{E}_y d(\mathcal{D}|y, \mathcal{D}_n|y). \quad (11)$$

The last term $\mathbb{E}_y d(\mathcal{D}|y, \mathcal{D}_n|y)$ is bounded by the proposed objective, i.e., Eq. 7. Thus, the transferred statistical robustness is bounded and the proof is complete.

$\square$

## A.2  PROOF FOR LEMMA A.1

*Proof.* To begin with, since the distance metric $d(\cdot, \cdot)$ is the Wasserstein distance, we have:

$$d(\mathcal{D}|y, \mathcal{D}_n|y) = \inf_{J \in \mathcal{J}(\mathcal{D}|y, \mathcal{D}_n|y)} \mathbb{E}_{(x, x') \sim J} m(x, x'), \quad (12)$$

where $\mathcal{J}(\mathcal{D}|y, \mathcal{D}_n|y)$ is the set of joint distributions. Let $\mathcal{J}^*$ be the optimal transport between $\mathcal{D}|y$ and $\mathcal{D}_n|y$. Then we have:

$$
\begin{aligned}
SR_d(f, \mathcal{D}(x,y)) &= \mathbb{E}_{(x,y)\sim\mathcal{D}} \inf_{f(x')\neq y} m\left(x', x\right) \\
&= \mathbb{E}_y \mathbb{E}_{x\sim\mathcal{D}|y} \inf_{f(x')\neq y} m\left(x', x\right) \\
&= \mathbb{E}_y \mathbb{E}_{(x,x'')\sim\mathcal{J}^*} \inf_{f(x')\neq y} m\left(x', x\right) \\
&\leq \mathbb{E}_y \mathbb{E}_{(x,x'')\sim\mathcal{J}^*} \inf_{f(x')\neq y} \left[m\left(x', x''\right) + m\left(x'', x\right)\right] \\
&= \mathbb{E}_y \mathbb{E}_{x''\sim\mathcal{D}_n|y} \inf_{f(x')\neq y} m\left(x'', x'\right) + \mathbb{E}_y \mathbb{E}_{(x,x'')\sim\mathcal{J}^*} m\left(x'', x\right) \\
&= \mathbb{E}_{(x'',y)\sim\mathcal{D}_n} \inf_{f(x')\neq y} m\left(x'', x'\right) + \mathbb{E}_y d\left(\mathcal{D}|y, \mathcal{D}_n|y\right) \\
&= SR_d\left(f, \mathcal{D}_n(x,y)\right) + \mathbb{E}_y d\left(\mathcal{D}|y, \mathcal{D}_n|y\right).
\end{aligned}
\tag{13}
$$

Similarly, we can also prove that:

$$
SR_d(f, \mathcal{D}_n(x,y)) \leq SR_d\left(f, \mathcal{D}(x,y)\right) + \mathbb{E}_y d\left(\mathcal{D}|y, \mathcal{D}_n|y\right).
\tag{14}
$$

Now using Eq. 13 and 14 we have:

$$
-\mathbb{E}_y d\left(\mathcal{D}|y, \mathcal{D}_n|y\right) \leq SR_d(f, \mathcal{D}) - SR_d\left(f, \mathcal{D}_n\right) \leq \mathbb{E}_y d\left(\mathcal{D}|y, \mathcal{D}_n|y\right).
\tag{15}
$$

Thus, we complete the proof.

$\square$

## B  MORE RELATED WORK

**Federated Learning.** FL is first proposed by McMahan et al. (2017) to protect data privacy in distributed machine learning. Training models within the FL framework can effectively safeguard privacy, as local data need not be shared. Instead of aggregating local data, the server aggregates local model updates from selected clients to update the global model in each round. To address specific problems within FL, various optimization algorithms have been proposed. FedCurv (Shoham et al., 2019) tackles the catastrophic forgetting problem of FL in the Non-IID case by drawing an analogy with lifelong learning. FedMA (Wang et al., 2020b) reduces the overall communication burden by constructing the global model in a layer-wise manner, matching and averaging hidden elements. There are also many algorithms proposed to address the issue of client drift (Li et al., 2020; Wang et al., 2020c; Karimireddy et al., 2020; Tang et al., 2022), such as FedNova (Wang et al., 2020c), which utilizes normalized averaging to eliminate objective inconsistency. VHL (Tang et al., 2022) also introduces surrogate data into FL, but they focus on solving data heterogeneity issues, while we focus on addressing backdoor attacks.

**Backdoor Attack on Federated Learning.** The goal of backdoor attacks is to modify the global model so that it can produce the desired target labels for inputs that possess specific properties (Shejwalkar et al., 2022). Bagdasaryan et al. (2020) investigates semantic backdoor attacks where the global model misclassifies input samples with the same semantic property, e.g. misclassifies the blue truck as a bird, and proposes a model-replacement attack to replace the global model. Bhagoji et al. (2019) discusses model poisoning attacks launched by a single malicious client. They boost the malicious updates to overcome the impact of updates from benign clients, and further propose alternating minimization and estimating benign updates to evade detection in almost every round. Wang et al. (2020a) proposes a new category of backdoor attacks called edge-case backdoors, and explains how these edge-case backdoors can lead to detection failures. Zhang et al. (2022) inserts more durable backdoors into FL systems by attacking parameters that are changed less in magnitude during training. Different from these works that only consider the centralized backdoor attack on FL, Xie et al. (2020) investigates the distributed backdoor attack (DBA), which decomposes a global trigger pattern into separate local patterns and embeds them into the training set of different adversarial parties respectively.

**Robust Federated Learning.** The goal of robust federated learning is to mitigate the impact of specific attacks during training. Blanchard et al. (2017) select model update(s) with the minimum squared distance to the updates of other clients. Coordinate-wise median (Yin et al., 2018) selects the median element coordinate-wise among the model updates of clients. Norm clipping (Sun et al., 2019) clips model updates whose norm exceeds a specific threshold. RFA (Pillutla et al., 2022) replaces the weighted arithmetic mean in FedAvg with a weighted geometric median, which is computed using the *smoothed Weiszfeld's algorithm*. FLTrust (Cao et al., 2020) mitigates the impact of backdoors by training models on the server-side with the additional root dataset, performing similarity checks based on the trained model and received local models. FLAME (Nguyen et al., 2022) eliminates backdoors by injecting noise into the model and employs HDBSCAN clustering and model weight clipping to reduce the required noise. FoolsGold (Fung et al., 2018) sums up the historical update vectors and calculates the cosine similarity between all participants to assign a global learning rate to each party. By giving lower learning rates to similar update vectors, Fools-Gold defends against label flipping and centralized backdoor attacks. SparseFed (Panda et al., 2022) utilizes global model top-k sparse updates and client-level gradient clipping to mitigate the impact of poisoning attacks. Our evaluation includes comparisons to six commonly used defense algorithms and demonstrates the stronger capabilities of SuDA against backdoor attacks.

## C  IMPLEMENTATION DETAILS

**Model replacement attack**: We form the backdoor task by conducting model replacement attacks (Bagdasaryan et al., 2020). In particular, the attacker trains the local model on the backdoored dataset and gets a backdoored model $X$. The attacker can arbitrarily manipulate the learning rate or training epochs to maximize the attack success rate on the backdoored data. In order to substitute the new global $G^{t+1}$ with the backdoored model $X$, the attacker scales up the weights of $X$ before sending it to the server:

$$L_{atk}^t = \gamma(X - G^t) + G^t,$$

where $\gamma$ is the scaling factor for the balance between attack capability and stealthiness. The scale factor we used is $\frac{m}{n_a}$, where $m$ is the number of clients participating in aggregation each round and $n_a$ is the number of attackers, which is consistent with previous outstanding works (Bagdasaryan et al., 2020; Wang et al., 2020a).

**Krum and Multi-Krum** (Blanchard et al., 2017): Given $n$ clients, Krum aims to defend against a maximum of $f$ attackers. In each round $r$, the server receives $n$ updates $(V_1^r, \cdots, V_n^r)$. For each update $V_i^r$, we denote $i \rightarrow j$ as the set of $n - f - 2$ closest updates to $V_i^r$. Then the score for each client $i$ is defined as the sum of squared distances between $V_i$ and each update $V_j$ in the set $i \rightarrow j$: $score(i) = \sum_{i \rightarrow j} \|V_i - V_j\|^2$. Krum then selects $V_{krum} = V_{i_*}$ with the lowest score $score(i_*) \leq score(i)$ for all $i$, and updates the global model as $w^{r+1} = w^r - V_{krum}$. While Multi-Krum selects $m \in \{1, \cdots, n\}$ updates $V_1^*, \cdots, V_m^*$ with the lowest scores, and calculates their average $\frac{1}{m} \sum_i V_i^*$ to replace $V_{krum}$. In our experiments, we apply $f = 6$ for both Krum and Multi-Krum and set $m = 8$ for Multi-Krum.

**Coordinate-wise median** (Yin et al., 2018): Given the set of updates $(V_1^r, \cdots, V_n^r)$ in each round, Coomed aggregates the updates: $\overline{V}^r = \text{Coomed}\{V_i^r : i \in [n]\}$, where the $j^{th}$ coordinate of $\overline{V}^r$ is given by $\overline{V}^r(j) = \text{med}\{V_i^r(j) : i \in [n]\}$. Here, the function med represents the 1-dimensional median, and $[n] = \{1, \cdots, n\}$.

**Norm clipping** (Sun et al., 2019): Due to the assumption that adversarial attacks can potentially generate updates with large norms, Normclip simply clips model updates whose norm exceeds a specific threshold $M$:

$$w_k^r = \frac{w_k^r}{max(1, \|w_k^r\|_2/M)}.$$

In our experiments, we set the threshold $M = 200$.

**RFA** (Pillutla et al., 2022): RFA replaces the weighted arithmetic mean utilized in FedAvg with a weighted geometric median:

$$\arg\min_v \sum_i \alpha_i \|v - w_i\|,$$

which is computed using the *smoothed Weiszfeld's algorithm*. The weight $\alpha_i$ is set to the proportion of training samples in the client $\alpha_i = \frac{n_i}{\sum_{j \in \mathcal{S}^r} n_j}$, where $\mathcal{S}^r$ is the subset of selected clients at round $r$. For iteration budget $R$ and the parameter $\nu$ in the smoothed Weiszfeld's algorithm, we set $R = 4$ and $\nu = 10^{-5}$.

**FLAME** (Nguyen et al., 2022): FLAME eliminates backdoors by injecting noise into the model and employs HDBSCAN clustering and model weight clipping to reduce the required noise. In our experiments, we set *min cluster size* to $(C \cdot K)/2 + 1$ and *min samples* to 1, which is consistent with the original paper.

**FLTrust** (Nguyen et al., 2022): FLTrust mitigates the impact of backdoors by training models on the server-side with the root dataset, performing similarity checks based on the trained model and received local models. We randomly collected 100 samples from the test set as the root dataset.

**SuDA**: SuDA generates the surrogate noise dataset using an un-pretrained StyleGAN-v2 (Karras et al., 2020). Clients then proceed to train local models with the SuDA objective parameter $\lambda$ set to 1, and the batch size is set to 128 for both real data and noise data. Then the server receives local models and reconstructs potential triggers. The sample number $b$ is set to 128. During the trigger optimization process, we gradually increase $\lambda$ to obtain as concise a trigger as possible while ensuring that the misclassification accuracy of the first term in Eq. 4 is greater than 98%. We set the number of noise samples $b$ used for reconstructing potential triggers to a fixed value of 128 for all experiments.

For all experiments, the learning rates are set to 0.01 and the learning rate decay is set to 0.992 per round. We employ momentum-SGD as optimizers, with momentum of 0.9 and weight decay of 0.0001. The degree of Non-IID local data distribution on the client is set to $\alpha = 1$. Our experiments were conducted on Ubuntu 20.04 LTS, Intel(R) Xeon(R) Platinum 8255C CPU, and 3090 GPU.

# D  ALGORITHM

In SuDA, the server generates the surrogate noise dataset at the beginning of the training phase and distributes the noise data to all clients. The clients proceed to train their respective local models using both the original dataset and the noise dataset, and send the trained local models back to the server. To effectively identify and mitigate backdoor attackers, the server reconstructs potential triggers for each local model, leveraging the presence of the noise data. We summarize the overall training procedure of SuDA in Algorithm 1.

In previous methods, malicious models are also involved in the calculation of these metrics, thus may yield tainted metrics and fail to achieve effective defense. For example, Krum (Blanchard et al., 2017) calculates the sum of the squared distances between each client model and the other client models as its score, and aggregates several models with the lowest scores. However, when the majority are attackers, the score of the malicious model may be relatively lower, leading the server to aggregate malicious models and resulting in defense failure. The proposed method aims to defend against backdoor attacks by designing a metric that will not be tainted by malicious models. To this end, we propose a metric that performs individual evaluation for each local model using surrogate data. This individual evaluation approach renders the metric impervious to variations in the attacker's ratio.

# E  MORE EXPERIMENTAL RESULTS

## E.1  DIFFERENT DATASETS AND RATIO

As shown in Tables 4 and 6, we conduct experiments on 3 datasets with different poison ratios, ranging from 1% to 20%. The number of attackers is 4. It can be seen that SuDA can effectively defend against backdoor attacks under different poison ratios. Note that although the accuracy of Normclip is sometimes slightly higher than SuDA, its Atk Rate is much higher in comparison. This is mainly because Normclip aggregates all clipped local models, which helps with model convergence but does not completely eliminate the negative impact caused by attackers. On the other hand, SuDA directly filters out malicious models by leveraging the surrogate dataset and does not select them in the aggregation process. Although this leads to a decrease in the number of clients participating

---

**Algorithm 1** Surrogate Data-guided Aggregation Strategy (SuDA)

---

**Input:** local epochs $E$, client number $K$, maximum round $R$, initial parameter $w^0$
**Output:** global parameter $w$
 **Initialization:** Server generates the surrogate noise dataset $\tilde{D}$, and distributes the initial model $w^0$ and $\tilde{D}$ to all clients.

 **Server:**
 **for** each round $r \in \{0, 1, \cdots, R\}$ **do**
  Uniformly selects a subset of clients $\mathcal{S}^r \subseteq \{1, \cdots, K\}$
  Sends the global model $w^r$ to all selected clients $k \in \mathcal{S}^r$
  **for** each client $k \in \mathcal{S}^r$ **in parallel do**
   $w_k^r \leftarrow \text{ClientTraining}(k, w^r)$
  **end for**
  $\mathcal{W}^r \leftarrow \{w_k^r | k \in \mathcal{S}^r\}$
  //Accuracy test
  $\mathcal{W}_1^r \leftarrow \text{AccFilter}(\mathcal{W}^r, \tilde{D})$
  //Potential Trigger Construction
  $\mathcal{T}^r \leftarrow \text{TriggerConstr}(\mathcal{W}_1^r, \tilde{D})$
  //Outlier Detection
  $\mathcal{W}_2^r \leftarrow \text{MAD}(\mathcal{T}^r)$
  //Aggregation
  $w^{r+1} \leftarrow \sum p_k w_k^r, w_k^r \in \mathcal{W}_2^r$
 **end for**

 **Benign Client:**
 **for** each epoch $e \in \{0, \cdots, E-1\}$ **do**
  $w_{k,e+1}^r \leftarrow w_{k,e}^r - \eta_{k,e} \nabla F_k^{SuDA}\left(w_{k,e}^r\right)$
 **end for**
 Return $w_k^r$ to sever

 **Compromised Client:**
 Injects the backdoor into the local dataset
 **for** each epoch $e \in \{0, \cdots, E-1\}$ **do**
  $w_{k,e+1}^r \leftarrow w_{k,e}^r - \eta_{k,e} \nabla F_k^{SuDA}\left(w_{k,e}^r\right)$
 **end for**
 $w_{atk}^r \leftarrow \gamma(w_k^r - w^r) + w^r$
 Return $w_{atk}^r$ to sever

---

in server aggregation, making it more challenging to converge, SuDA still achieves a significantly lower Atk Rate than Normclip while maintaining comparable Acc and Main Acc.

E.2    EXPERIMENTS WITH FEDPROX

FedProx (Li et al., 2020) is one of the more common training methods than FedAvg when extreme heterogeneity exists in the client data. Therefore, we conduct experiments with FedProx on CIFAR-10 in the Non-IID setting. We set the Non-IID degree control parameter $\alpha = 1$ and the poison ratio is 5%. The experimental results are shown in Table 5. It can be seen that SuDA is easy to integrate with FedProx and performs well against backdoor attacks.

E.3    PERFORMANCE IN THE IID SETTING

To further demonstrate the applicability of SuDA, we compared the performance of defense methods on 3 datasets in the IID setting. We set the Non-IID degree control parameter $\alpha = 100$ to simulate the IID setting. The experimental results are shown in Tables 7. It can be seen that SuDA

Table 4: ACC, ASR and MA of defense algorithms on the dataset CIFAR-10 with different poison ratios.

| Poison Ratio | Defense | 4 attackers | | | 8 attackers | | | 12 attackers | | |
|---|---|---|---|---|---|---|---|---|---|---|
| | | ACC(%) | ASR(%) | MA(%) | ACC(%) | ASR(%) | MA(%) | ACC(%) | ASR(%) | MA(%) |
| 1% | FedAvg | 81.12 | 2.41 | 81.21 | 84.43 | 2.41 | 84.51 | 84.97 | 2.76 | 85.07 |
| | RFA | 84.62 | 1.55 | 84.72 | 85.37 | 1.77 | **85.47** | 85.13 | 1.79 | 85.22 |
| | Krum | 49.58 | 5.5 | 49.41 | 51.53 | 1.86 | 51.16 | 52.09 | 3.88 | 51.87 |
| | MultiKrum | 80.5 | 3.24 | 80.67 | 80.49 | 2.3 | 80.66 | 80.49 | 1.53 | 80.5 |
| | Coomed | 83.76 | 1.42 | 83.73 | 83.51 | 1.29 | 83.49 | 83.49 | 1.42 | 83.57 |
| | Normclip | 85.29 | 1.53 | 85.26 | 85.41 | 1.46 | 85.45 | **85.3** | 1.77 | **85.41** |
| | SuDA(ours) | **85.59** | **1.2** | **85.71** | 84.89 | **1.27** | 85.03 | 85.14 | **1.35** | 85.18 |
| 5% | FedAvg | 76.79 | 87.63 | 83.4 | 77.86 | 88.9 | 84.67 | 77.8 | 89.64 | 84.67 |
| | RFA | 79.38 | 76.31 | 85.2 | 78.69 | 83.42 | 85.07 | 77.9 | 87.76 | 84.61 |
| | Krum | 49.58 | 5.5 | 49.41 | 50.91 | 6.44 | 50.79 | 48.03 | 8.47 | 47.9 |
| | MultiKrum | 80.5 | 3.24 | 80.67 | 78.73 | 6.28 | 79 | 76.99 | 10.98 | 77.72 |
| | Coomed | 76.75 | 58.44 | 80.9 | 77.27 | 81.88 | 83.42 | 77.12 | 84.62 | 83.49 |
| | Normclip | 79.41 | 78.17 | **85.41** | 78.85 | 83.33 | 85.26 | 78.68 | 87.91 | **85.47** |
| | SuDA(ours) | 85.2 | 2.3 | 85.31 | **85.25** | 2.49 | 85.42 | 84.68 | 2.87 | 84.9 |
| 10% | FedAvg | 76.62 | 92.21 | 83.61 | 78.09 | 87.93 | 84.94 | 77.88 | 91.84 | 84.94 |
| | RFA | 78.44 | 84.71 | 84.95 | 78.09 | 88.96 | 84.94 | 78.18 | 91.31 | 85.22 |
| | Krum | 49.58 | 5.5 | 49.41 | 51.89 | 3.83 | 51.55 | 51.64 | 2.03 | 51.28 |
| | MultiKrum | 80.5 | 3.24 | 80.67 | 79.18 | 3 | 79.26 | 74.45 | 50.3 | 77.79 |
| | Coomed | 77.51 | 81.64 | 83.67 | 76.74 | 86.9 | 83.28 | 77.11 | 88.99 | 83.85 |
| | Normclip | 79.04 | 86.05 | **85.7** | 78.58 | 87.52 | **85.35** | 78.55 | 90.61 | **85.55** |
| | SuDA(ours) | **85.38** | **2.06** | 85.52 | **85.07** | **2.03** | 85.2 | **84.79** | **2.03** | 84.87 |
| 20% | FedAvg | 73.65 | 91.92 | 80.34 | 78.21 | 90.72 | 85.23 | 76.86 | 92.58 | 83.89 |
| | RFA | 76.47 | 87.71 | 83.08 | 77.83 | 85.65 | 84.39 | 75.62 | 85.46 | 82.01 |
| | Krum | 49.58 | 5.5 | 49.41 | 52.07 | 2.3 | 51.63 | 46.69 | 3.2 | 46.5 |
| | MultiKrum | 80.5 | 3.24 | 80.67 | 79.27 | 1.6 | 79.36 | 75.14 | 68.37 | 79.98 |
| | Coomed | 77.03 | 86.31 | 83.54 | 76.65 | 88.77 | 83.31 | 76.87 | 89.84 | 83.68 |
| | Normclip | 78.62 | 88.81 | 85.49 | 78.7 | 89.05 | **85.62** | 78.32 | 91 | **85.37** |
| | SuDA(ours) | **85.53** | **2.01** | **85.71** | **84.92** | **1.51** | 84.97 | **84.28** | **2.01** | 84.42 |

Table 5: Performance of different defense algorithms with FedProx on CIFAR-10.

| Defense | Atk Num = 0 | | | Atk Num = 4 | | | Atk Num = 8 | | | Atk Num = 12 | | |
|---|---|---|---|---|---|---|---|---|---|---|---|---|
| | ACC(%) | ASR(%) | MA(%) | ACC(%) | ASR(%) | MA(%) | ACC(%) | ASR(%) | MA(%) | ACC(%) | ASR(%) | MA(%) |
| FedAvg | 84.92 | 1.97 | 85.03 | 75.47 | 89.56 | 82.13 | 77.61 | 88.9 | 84.39 | 78.34 | 88.9 | **85.21** |
| RFA | 84.27 | 2.23 | 84.4 | 79.46 | 77.67 | 85.36 | 78.71 | 83.26 | **85.08** | 77.78 | 88.07 | 84.5 |
| Krum | 54.54 | 2.82 | 54.51 | 54.64 | 4.34 | 54.51 | 53.53 | 6.44 | 53.45 | 53.41 | 8.17 | 53.38 |
| MultiKrum | 80.71 | 1.73 | 80.84 | 79.58 | 2.03 | 79.66 | 78.87 | 2.22 | 79.04 | 78.21 | 28.44 | 80.23 |
| Coomed | 84.04 | 1.44 | 84.06 | 78.14 | 65.96 | 82.96 | 77.38 | 81.71 | 83.49 | 77.18 | 85.41 | 83.61 |
| Normclip | 84.71 | **1.42** | 84.72 | 78.71 | 77.19 | 84.55 | 78.29 | 83.35 | 84.65 | 78.3 | 86.82 | 84.96 |
| SuDA(ours) | **86.08** | 1.53 | **86.17** | **85.42** | **1.82** | **85.51** | **84.48** | **1.99** | 84.61 | **84.67** | **2.1** | 84.79 |

can still effectively defend against backdoor attacks in the IID setting and preserve high accuracy simultaneously.

### E.4 SURROGATE DATASET GENERATION

To further investigate the effect of different surrogate datasets, we employ two additional generated datasets to replace the original surrogate dataset produced by StyleGAN. These two datasets include one generated by a simple CNN and another generated by upsampling pure Gaussian noise. In our data generation methods, we sample noise from various Gaussian distributions, each with the same mean but different standard deviations, to generate noise with diverse latent styles that correspond to distinct classes. Given that datasets CIFAR-10, FMNIST and SVHN each consist of 10 classes, the surrogate dataset also comprises 10 classes. The size of the surrogate dataset is 2000 in our experiments, which is a small proportion of the utilized datasets, i.e., 3.33% for CIFAR-10, 2.86% for FMNIST, and 0.33% for SVHN. We show the generated surrogate datasets as Figures 4, 5 and 6. We also conduct ablations on the sensitivity of the size of the surrogate dataset and report results in Table 10.

For the dataset generated by the simple CNN, we first sample 64-dimensional noises. These noises are then fed into a CNN composed of 4 transpose convolutional layers and 3 convolutional layers. The CNN model processes the input and produces noise data of size $32 \times 32$. We employ 10 CNNs

Table 6: ACC, ASR and MA of defense algorithms on CIFAR-10, FMNIST and SVHN with different poison ratios.

| Poison Ratio | Defense | CIFAR-10 | | | FMNIST | | | SVHN | | |
|---|---|---|---|---|---|---|---|---|---|---|
| | | ACC(%) | ASR(%) | MA(%) | ACC(%) | ASR(%) | MA(%) | ACC(%) | ASR(%) | MA(%) |
| 1% | FedAvg | 81.12 | 2.41 | 81.21 | 90.5 | 8.57 | 91.21 | 87.66 | 2.92 | 87.79 |
| | RFA | 84.62 | 1.55 | 84.72 | 91.49 | 1.9 | 91.59 | 89.25 | 1.23 | 89.32 |
| | Krum | 49.58 | 5.5 | 49.41 | 84.12 | 5.3 | 84.23 | 83.51 | 1.26 | 83.55 |
| | MultiKrum | 80.5 | 3.24 | 80.67 | 91.23 | 1.99 | 91.36 | 86.47 | 1.37 | 86.53 |
| | Coomed | 83.76 | 1.42 | 83.73 | **92** | 1.77 | **92.17** | 89.16 | 1.28 | 89.23 |
| | Normclip | 85.29 | 1.53 | 85.26 | 91.67 | 1.6 | 91.83 | 87.74 | 1.46 | 87.82 |
| | SuDA(ours) | **85.59** | **1.2** | **85.71** | 91.36 | **1.6** | 91.43 | **90.45** | **0.99** | **90.52** |
| 5% | FedAvg | 76.79 | 87.63 | 83.4 | 83.71 | 99.78 | **92.09** | 81.41 | 67.14 | 86.66 |
| | RFA | 79.38 | 76.31 | 85.2 | 87.41 | 22.74 | 89.2 | 88.99 | 5.36 | 89.36 |
| | Krum | 49.58 | 5.5 | 49.41 | 84.12 | 5.3 | 84.23 | 83.51 | 1.26 | 83.55 |
| | MultiKrum | 80.5 | 3.24 | 80.67 | 91.23 | 1.99 | 91.36 | 86.47 | 1.37 | 86.53 |
| | Coomed | 76.75 | 58.44 | 80.9 | 88.93 | 5.83 | 89.36 | 89.04 | 3.09 | 89.25 |
| | Normclip | 79.41 | 78.17 | **85.41** | 88.94 | 8.55 | 89.58 | 87.58 | 2.19 | 87.7 |
| | SuDA(ours) | **85.2** | **2.3** | 85.31 | **91.46** | **1.44** | 91.55 | **90.72** | **1** | **90.75** |
| 10% | FedAvg | 76.62 | 92.21 | 83.61 | 83.46 | 99.75 | **91.82** | 79.6 | 89.29 | 86.72 |
| | RFA | 78.44 | 84.71 | 84.95 | 85.4 | 28.68 | 87.63 | 87.34 | 10.51 | 88.05 |
| | Krum | 49.58 | 5.5 | 49.41 | 84.12 | 5.3 | 84.23 | 83.51 | 1.26 | 83.55 |
| | MultiKrum | 80.5 | 3.24 | 80.67 | 91.23 | 1.99 | 91.36 | 86.47 | 1.37 | 86.53 |
| | Coomed | 77.51 | 81.64 | 83.67 | 87.42 | 11.29 | 88.27 | 87.96 | 13.88 | 88.93 |
| | Normclip | 79.04 | 86.05 | **85.7** | 86.83 | 12.93 | 87.77 | 87.3 | 5.74 | 87.69 |
| | SuDA(ours) | **85.38** | **2.06** | 85.52 | **91.32** | **1.57** | 91.37 | **90.6** | **0.82** | **90.67** |
| 20% | FedAvg | 73.65 | 91.92 | 80.34 | 83.19 | 99.78 | **91.53** | 78.39 | 95.28 | 85.95 |
| | RFA | 76.47 | 87.71 | 83.08 | 80.03 | 34.62 | 82.38 | 84.08 | 35.51 | 86.73 |
| | Krum | 49.58 | 5.5 | 49.41 | 84.12 | 5.3 | 84.23 | 83.51 | 1.26 | 83.55 |
| | MultiKrum | 80.5 | 3.24 | 80.67 | **91.23** | 1.99 | 91.36 | 86.47 | 1.37 | 86.53 |
| | Coomed | 77.03 | 86.31 | 83.54 | 85.88 | 21.77 | 87.49 | 85.84 | 17.67 | 87.08 |
| | Normclip | 78.62 | 88.81 | 85.49 | 84.91 | 18.92 | 86.32 | 84.42 | 22.04 | 86.01 |
| | SuDA(ours) | **85.53** | **2.01** | **85.71** | 90.91 | **1.64** | 90.95 | **91** | **0.76** | **91.05** |

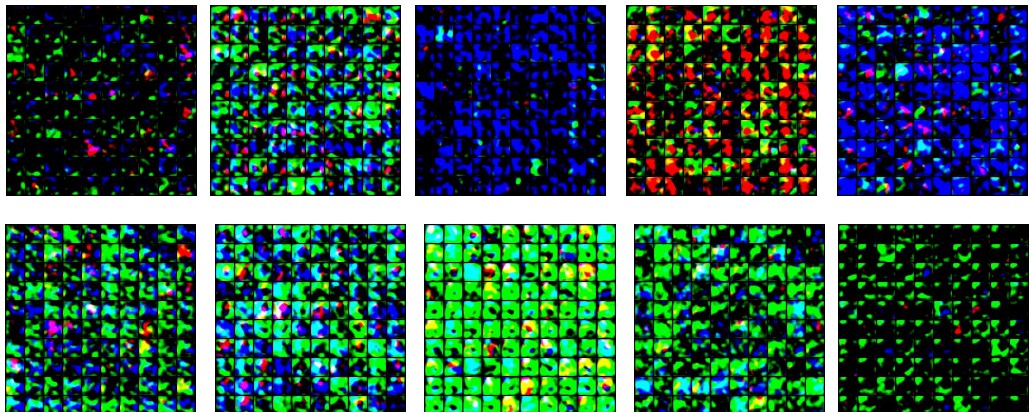

Figure 4: Surrogate dataset generated by the StyleGAN.

with distinct initial weights to generate the noise data that have enough diversity as the dataset of 10 classes.

For the dataset generated by upsampling the pure Gaussian noise, noise points are initially sampled to form an image of size $8 \times 8$. Subsequently, upsampling is employed to transform the image into a larger size of $32 \times 32$. This upsampling process enables the generation of noise images with some low-level features, thereby enabling the model to learn basic feature distributions from them.

As shown in Table 8, surrogate datasets generated by these two methods can also provide powerful defense capabilities to the server, which further demonstrates the applicability and relevance of SuDA.

Table 7: Performance of defense algorithms on CIFAR-10, FMNIST and SVHN when defending against varying attackers in the IID setting.

| Atk Num | Defense | CIFAR-10 | | | FMNIST | | | SVHN | | |
|---|---|---|---|---|---|---|---|---|---|---|
| | | ACC(%) | ASR(%) | MA(%) | ACC(%) | ASR(%) | MA(%) | ACC(%) | ASR(%) | MA(%) |
| 0 | FedAvg | **79** | 2.32 | **78.89** | 91.99 | 1.86 | **92.09** | **87.63** | 1.54 | **87.99** |
| | RFA | 78.65 | 2.34 | 78.55 | 91.51 | 1.97 | 91.61 | 87.49 | 1.48 | 87.81 |
| | Krum | 61.73 | 7.39 | 61.62 | 87.2 | 2.17 | 87.19 | 79.89 | 3.05 | 80.27 |
| | MultiKrum | 76.49 | 2.21 | 76.37 | 90.7 | 1.99 | 90.78 | 85.62 | 2.17 | 85.94 |
| | Coomed | 78.35 | 2.14 | 78.2 | **92.01** | 1.79 | 92.09 | 87.56 | 1.55 | 87.76 |
| | Normclip | 77 | 2.47 | 76.87 | 91.65 | 1.84 | 91.7 | 82.6 | 2.52 | 82.78 |
| | SuDA(ours) | 78.98 | **1.46** | 78.8 | 90.97 | **1.55** | 91 | 86.44 | **1.01** | 86.5 |
| 4 | FedAvg | 74.32 | 62.6 | 78.46 | 83.81 | 95.65 | **91.84** | 80.08 | 97.08 | **87.99** |
| | RFA | 74.86 | 60.96 | **78.89** | 88.71 | 19.6 | 90.32 | 80.21 | 96.68 | 88.1 |
| | Krum | 58.42 | 6.38 | 58.37 | 87.52 | 2.63 | 87.47 | 81.11 | 2.66 | 81.44 |
| | MultiKrum | 76.13 | 2.55 | 76.11 | 90.77 | 2.25 | 90.89 | 85.63 | 1.8 | 85.98 |
| | Coomed | 76.82 | 31.82 | 78.63 | 89.44 | 19.03 | 90.82 | 79.37 | 73.9 | 85.11 |
| | Normclip | 73.15 | 59.03 | 76.86 | 89.89 | 5.3 | 90.27 | 82.63 | 3 | 82.84 |
| | SuDA(ours) | **78.57** | **1.75** | 78.37 | **91.33** | **1.64** | 91.41 | **85.91** | **1.06** | 86.05 |
| 8 | FedAvg | 73.38 | 77.93 | 78.76 | 83.58 | 98.24 | **91.83** | 79.7 | 98.85 | **87.73** |
| | RFA | 73.52 | 76.79 | **78.83** | 86.74 | 29.45 | 89.08 | 79.67 | 98.88 | 88.71 |
| | Krum | 59.75 | 6.09 | 59.73 | 85.88 | 2.74 | 86.03 | 80 | 4.25 | 81.44 |
| | MultiKrum | 76.12 | 4.16 | 76.1 | **90.91** | 2.54 | 91.04 | **85.01** | 5.43 | 85.67 |
| | Coomed | 73.29 | 73.28 | 78.28 | 86.76 | 31.88 | 89.3 | 79.77 | 98.43 | 87.77 |
| | Normclip | 71.93 | 73.48 | 76.78 | 87.81 | 19.36 | 89.3 | 82.51 | 3.23 | 82.71 |
| | SuDA(ours) | **78.32** | **2.19** | 78.21 | 90.68 | **2.02** | 90.87 | 84.48 | **1.18** | 84.64 |
| 12 | FedAvg | 73.21 | 81.77 | **78.94** | 83.63 | 99.07 | **91.95** | 79.82 | 99.55 | **87.94** |
| | RFA | 72.95 | 81.2 | 78.63 | 84.16 | 59.6 | 89.01 | 79.57 | 99.49 | 87.65 |
| | Krum | 60.76 | 7.41 | 60.73 | 86.4 | 4.16 | 86.48 | 76.34 | 64.06 | 81.04 |
| | MultiKrum | 76.14 | 5.29 | 76.25 | 89.88 | 5.41 | 90.22 | 77.72 | 96.6 | 85.35 |
| | Coomed | 73.24 | 80.08 | 78.83 | 84.42 | 56.46 | 89.03 | 79.59 | 99.33 | 87.66 |
| | Normclip | 71.49 | 79.38 | 76.87 | 85.03 | 52.32 | 89.27 | 82.28 | 3.92 | 82.52 |
| | SuDA(ours) | **77.55** | **2.41** | 77.38 | **90.06** | **3.13** | 90.19 | **85.19** | **1.61** | 85.35 |

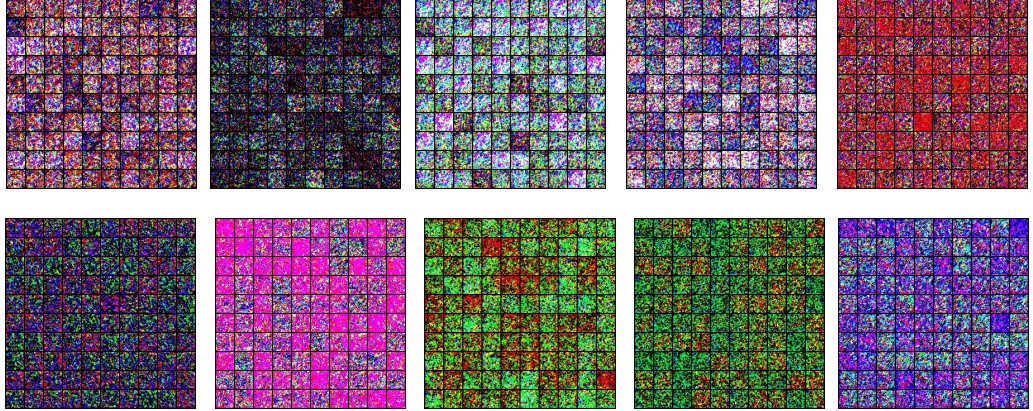

Figure 5: Surrogate dataset generated by the simple CNN.

## E.5 DIFFERENT GLOBAL MODEL

We investigate the sensitivity of SuDA to various shared global models. In particular, we conduct experiments on CIFAR-10 to compare the performance of SuDA with different defense algorithms on a range of models, including ResNet-10, ResNet-34 (He et al., 2016), VGG-9 and VGG-19 (Simonyan & Zisserman, 2014). The results presented in Table 9 demonstrate the effectiveness of SuDA across models of varying capacities.

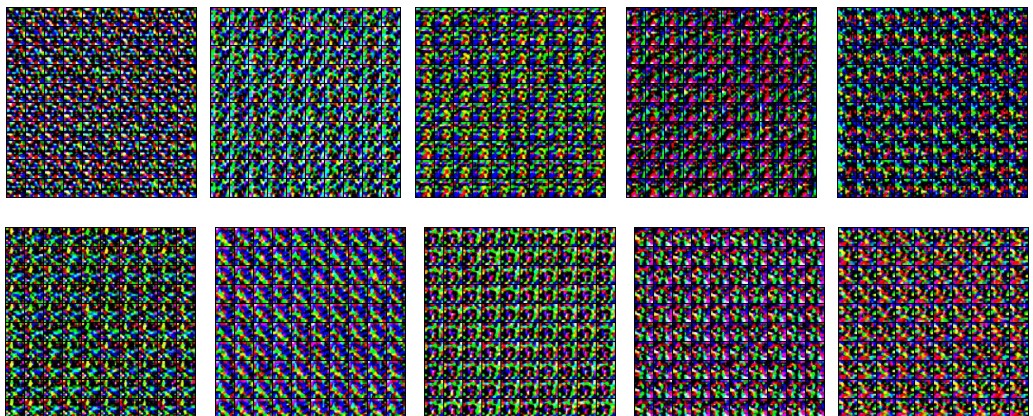

Figure 6: Surrogate dataset generated by upsampling the pure Gaussian noise.

Table 8: Results of SuDA with different generated noise datasets on CIFAR-10, FMNIST and SVHN.

| Atk Num | Defense | CIFAR-10 | | | FMNIST | | | SVHN | | |
|---|---|---|---|---|---|---|---|---|---|---|
| | | ACC(%) | ASR(%) | MA(%) | ACC(%) | ASR(%) | MA(%) | ACC(%) | ASR(%) | MA(%) |
| 0 | FedAvg | 84.72 | 3.26 | 84.87 | **91.51** | 1.77 | **91.64** | 89.2 | 1.12 | 89.28 |
| | SuDA | **86.29** | **1.33** | **86.28** | 91.2 | **1.38** | 91.26 | **90.32** | **0.82** | **90.39** |
| | SuDA Gaus | 84.61 | 1.66 | 84.71 | 91.09 | 1.71 | 91.21 | 90.09 | 0.83 | 90.15 |
| | SuDA CNN | 84.01 | 1.51 | 84.02 | 90.78 | 1.71 | 90.91 | 89.97 | 1.25 | 90.05 |
| 4 | FedAvg | 76.79 | 87.63 | 83.4 | 83.71 | 99.78 | 92.09 | 81.41 | 67.14 | 86.66 |
| | SuDA | **85.2** | **2.3** | **85.31** | **91.46** | **1.44** | **91.55** | 90.72 | 1 | 90.75 |
| | SuDA Gaus | 81.67 | 2.43 | 81.82 | 90.49 | 1.9 | 90.62 | **90.88** | **0.85** | **90.95** |
| | SuDA CNN | 81.82 | 2.87 | 81.88 | 90.31 | 2.03 | 90.37 | 89.18 | 1.4 | 89.27 |
| 8 | FedAvg | 77.86 | 88.9 | 84.67 | 83.27 | 99.67 | **91.61** | 80.79 | 66.21 | 85.9 |
| | SuDA | **85.25** | 2.49 | **85.42** | 90.79 | 1.82 | 90.89 | **90.68** | **0.88** | **90.72** |
| | SuDA Gaus | 83.16 | **1.46** | 83.22 | **90.84** | 1.88 | 90.97 | 89.74 | 3.18 | 89.9 |
| | SuDA CNN | 80.47 | 1.6 | 80.49 | 90.37 | **1.55** | 90.45 | 89.68 | 3.36 | 89.87 |
| 12 | FedAvg | 77.8 | 89.64 | 84.67 | 83.34 | 99.64 | **91.68** | 81.58 | 75.29 | 87.61 |
| | SuDA | **84.68** | 2.87 | **84.9** | **87.29** | **3.07** | 87.38 | **89.48** | **1.44** | **89.52** |
| | SuDA Gaus | 84.41 | **2.23** | 84.49 | 85.54 | 3.37 | 85.64 | 87.19 | 3.8 | 87.41 |
| | SuDA CNN | 80.97 | 2.67 | 80.96 | 85.99 | 6.31 | 86.37 | 85.45 | 4.56 | 85.71 |

### E.6 DIFFERENT HYPERPARAMETER $\lambda$

We adjust the align weight $\lambda$ in the SuDA objective $F_k^{SuDA}$ from 0.1 to 5 on CIFAR-10 to examine the sensitivity of SuDA to $\lambda$. The results in Table 15 demonstrate that SuDA is not sensitive to the align weight $\lambda$, and it can achieve good performance within a wide range of $\lambda$.

### E.7 MORE ADAPTIVE ATTACK SCENARIOS

In Table 2, we show the results of SuDA against the adaptive attack scenario: the attacker changes the alignment parameter $\lambda$ used to a different one not used by the benign clients. Empirical results show that SuDA still performs relatively well against such an adaptive attack. We also consider the performance of SuDA under another adaptive attack scenario: the attackers divide their poisoned dataset into poisoned and benign parts, and only align the surrogate samples with the data within the benign parts. The experimental results are shown in Tables 16 and 17. It can be seen that although SuDA's performance is slightly worse under this adaptive attack, it is still better than other defense methods.

To further demonstrate the effectiveness of the proposed method, we consider stronger attackers, pixel-level triggers (Doan et al., 2021) and distributed backdoor attacks (DBA) (Xie et al., 2020). The experimental results in Table 12 show that SuDA can effectively defend against these attacks.

Table 9: Performance of different defense algorithms on different models on CIFAR-10.

| Defense | ResNet-10 | | | ResNet-34 | | | VGG-9 | | | VGG-19 | | |
|---|---|---|---|---|---|---|---|---|---|---|---|---|
| | ACC(%) | ASR(%) | MA(%) | ACC(%) | ASR(%) | MA(%) | ACC(%) | ASR(%) | MA(%) | ACC(%) | ASR(%) | MA(%) |
| FedAvg | 72.07 | 85.32 | 78.09 | 73.36 | 88.13 | 79.69 | 45 | 8.57 | 44.87 | 45.33 | 9.42 | 45.28 |
| RFA | 76.68 | 73.75 | **82** | 76.88 | 71.42 | 82.09 | 45.53 | 6.09 | 45.22 | 41.53 | 7.19 | 41.44 |
| Krum | 53.44 | 6.33 | 53.36 | 43.42 | 7.63 | 43.24 | 26 | 29.21 | 26.39 | 18.44 | 45.17 | 19.01 |
| MultiKrum | 75.04 | 2.47 | 75 | 80.46 | 3.33 | 80.59 | 43.04 | 6.95 | 42.82 | 43.7 | 10.13 | 43.62 |
| Coomed | 76.18 | 62.25 | 80.49 | 79.11 | 65.28 | 84 | 32.2 | 5.98 | 32.1 | 23.37 | 3.2 | 23.34 |
| Normclip | 75.13 | 66 | 79.7 | 78.39 | 77.12 | 84.19 | 45.96 | 4.64 | 45.57 | 42.55 | 7.08 | 42.46 |
| SuDA(ours) | **80.28** | **1.73** | 80.46 | **86.53** | **1.71** | **86.91** | **56.63** | **1.58** | **56.95** | **55.1** | **2.52** | **55.91** |

Table 10: Results of SuDA with different surrogate dataset size.

| Defense | Metric | Surrogate Dataset Size | | | | | | |
|---|---|---|---|---|---|---|---|---|
| | | 500 | 1000 | 2000 | 3000 | 4000 | 6000 | 8000 |
| SuDA | ACC(%) | 87.74 | **87.89** | 87.67 | 87.26 | 86.69 | 87.75 | 87.61 |
| | ASR(%) | 1.6 | 3.22 | 2.23 | **1.18** | 3.17 | 2.23 | 3.26 |
| | MA(%) | 87.84 | **88.01** | 87.88 | 87.27 | 86.89 | 87.87 | 87.81 |

Table 11: Results of SuDA with different noise sample number.

| Defense | Metric | Noise Sample Num $b$ | | | | |
|---|---|---|---|---|---|---|
| | | 64 | 128 | 256 | 512 | 1024 |
| SuDA | ACC(%) | 87.31 | 87.67 | **87.7** | 87.54 | 87.24 |
| | ASR(%) | 1.92 | 2.23 | 2.89 | **1.9** | 2.69 |
| | MA(%) | 87.42 | **87.88** | 87.83 | 87.69 | 87.49 |

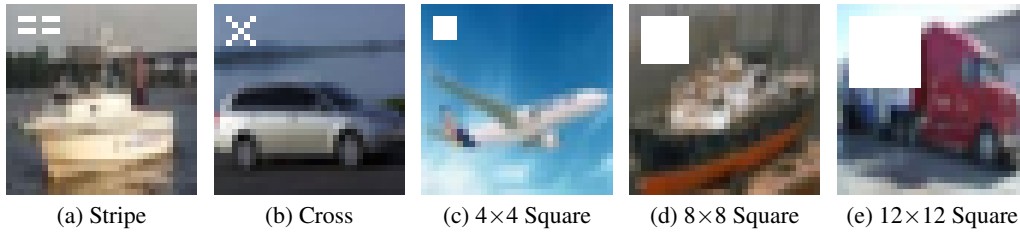

| (a) Stripe | (b) Cross | (c) 4×4 Square | (d) 8×8 Square | (e) 12×12 Square |

Figure 7: Different Trigger Patterns

This is intuitive because these attacks do not fundamentally change how the triggers are injected, allowing SuDA to reconstruct the triggers through optimization.

### E.8 DIFFERENT TRIGGER PATTERNS

We investigate the defensive capabilities of SuDA against different trigger patterns. As shown in Figure 7, we investigate two additional, more complex triggers, as well as larger-sized triggers. The experimental results are shown in Table 13. It can be seen that SuDA performs well against two more complex triggers, 'Stripe' and 'Cross'. When defending against large triggers, SuDA still performs well in defending against triggers with a size of 10, but if the triggers continue to increase in size, the defense performance will decrease.

### E.9 EXTRA TIME OVERHEAD

In order to minimize the extra time overhead caused by reconstructing potential triggers, we record the optimized trigger obtained in each round as the initial value for the next round of optimization, thus eliminating the need to start the estimation from scratch in each round. At the same time, we use early-stop during trigger optimization, which greatly reduces the time overhead after the 60th round. To further reduce time overhead, we also investigate a more efficient method, SuDA-Efficient. SuDA-Efficient does not reconstruct potential triggers for each label of each local model sequentially; instead, it first aggregates all local models, reconstructs potential triggers for the ag-

Table 12: Results of stronger attackers.

| Defense | CIFAR-10 | | | FMNIST | | |
|---|---|---|---|---|---|---|
| | ACC(%) | ASR(%) | MA(%) | ACC(%) | ASR(%) | MA(%) |
| Replacement Attack | 84.65 | 2.38 | 84.77 | **90.80** | 1.20 | **90.92** |
| Pixel-level Triggers Attack | 84.17 | 3.57 | 84.40 | 90.54 | **1.18** | 90.80 |
| DBA | **84.92** | **2.29** | **85.39** | 90.29 | 1.34 | 90.62 |

Table 13: Results of SuDA with different Trigger Patterns.

| Defense | Metric | Stripe | Cross | Square Size | | | | | |
|---|---|---|---|---|---|---|---|---|---|
| | | | | 4 | 6 | 8 | 10 | 12 | 14 |
| SuDA | ACC(%) | 84.80 | 84.95 | 84.65 | 83.84 | 83.43 | 83.96 | 82.13 | 77.15 |
| | ASR(%) | 1.09 | 2.52 | 2.38 | 1.22 | 6.22 | 2.61 | 15.32 | 89.07 |
| | MA(%) | 84.91 | 85.10 | 84.77 | 83.94 | 84.17 | 84.64 | 83.97 | 83.92 |

Table 14: Comparison of extra time overhead between SuDA and baseline methods

| Defense | sec per Round | ACC(%) | ASR(%) | MA(%) |
|---|---|---|---|---|
| FedAvg | 28.67 | 76.79 | 87.63 | 83.40 |
| RFA | 28.84 | 79.38 | 76.31 | 85.20 |
| Krum | 30.90 | 49.58 | 5.50 | 49.41 |
| Coomed | 29.15 | 76.75 | 58.44 | 80.90 |
| Normclip | 32.08 | 79.41 | 78.17 | **85.41** |
| FLAME | 33.13 | 77.19 | 82.65 | 83.36 |
| FLTrust | 34.50 | 65.54 | 38.99 | 67.57 |
| SuDA(ours) | 61.71 | **84.65** | **2.38** | 84.77 |
| SuDA-Efficient(ours) | 38.42 | 82.78 | 8.22 | 83.35 |

Table 15: Performance of SuDA on CIFAR-10 for varying align parameter $\lambda$.

| $\lambda$ | Atk Num = 0 | | | Atk Num = 4 | | | Atk Num = 8 | | | Atk Num = 12 | | |
|---|---|---|---|---|---|---|---|---|---|---|---|---|
| | ACC(%) | ASR(%) | MA(%) | ACC(%) | ASR(%) | MA(%) | ACC(%) | ASR(%) | MA(%) | ACC(%) | ASR(%) | MA(%) |
| $\lambda = 0.1$ | 83.94 | 2.76 | 84.03 | 83.11 | 2.93 | 83.23 | 82.84 | 2.01 | 82.94 | 82.22 | 2.85 | 82.34 |
| $\lambda = 0.2$ | 84.45 | 2.25 | 84.56 | 83.66 | 3.09 | 83.82 | 83.33 | 2.23 | 83.44 | 83.01 | 3.24 | 83.16 |
| $\lambda = 0.5$ | 85.96 | 1.86 | 85.98 | 84.68 | 2.57 | 84.8 | 84.24 | 2.27 | 84.39 | 84.08 | 3.05 | 84.25 |
| $\lambda = 1$ | 86.29 | **1.33** | 86.28 | 85.2 | 2.3 | 85.31 | 85.25 | 2.49 | 85.42 | 84.68 | 2.87 | 84.9 |
| $\lambda = 2$ | 88.12 | 1.67 | 88.32 | 85.71 | **1.61** | 85.74 | 85.51 | 2.37 | 85.66 | 85.37 | 2.24 | 85.46 |
| $\lambda = 5$ | **88.58** | 1.82 | **88.73** | **85.87** | 2.12 | **86.06** | 85.87 | **1.46** | **85.98** | **86.22** | **1.49** | **86.37** |

gregated model, and checks whether the target label exists. When the target label is detected, it then reconstructs potential triggers for that label in each local model, filtering out malicious clients. When $K$ clients are involved in the aggregation, SuDA-Efficient can reduce the extra time overhead by up to $K$ times.

We conduct experiments to compare SuDA and baseline methods on CIFAR-10. The experimental results are shown in Table 14. From the table, we can see that SuDA achieves excellent defense performance, while keeping the time overhead acceptable. SuDA-Efficient further reduces the time overhead and still maintains good defense performance compared to other baseline methods.

### E.10 MORE ABLATION EXPERIMENTS

To further demonstrate the effectiveness of SuDA, we conduct more ablation experiments on CIFAR-10. We consider a new scenario: a total of 50 clients, all participating in aggregation each round. Tables 18 and 19 show the performance of different defense algorithms under large attacker ratios and large poison ratios respectively, which further underscore the robust effectiveness of SuDA in diverse settings.

In Table 11, we conduct ablations on the sensitivity of b mentioned in Section 4.2. We can see that b has a limited impact on the performance. In Table 10, we investigate the effect of the size of the surrogate dataset on the performance. We can see that the size of the surrogate dataset has a limited impact on the performance. Even when the surrogate dataset size is only 500, the proposed method still demonstrates excellent performance.

Table 16: Performance of SuDA under the adaptive attack when defending against varying attackers.

| Atk Num | Defense | CIFAR-10 | | | FMNIST | | | SVHN | | |
|---|---|---|---|---|---|---|---|---|---|---|
| | | ACC(%) | ASR(%) | MA(%) | ACC(%) | ASR(%) | MA(%) | ACC(%) | ASR(%) | MA(%) |
| 0 | SuDA | **86.29** | **1.33** | **86.28** | **91.2** | **1.38** | **91.26** | **90.32** | **0.82** | **90.39** |
| | SuDA Adapt | 86.29 | 1.33 | 86.28 | 91.2 | 1.38 | 91.26 | 90.32 | 0.82 | 90.39 |
| 4 | SuDA | **85.2** | **2.3** | **85.31** | **91.46** | **1.44** | **91.55** | 90.72 | 1 | 90.75 |
| | SuDA Adapt | 84.28 | 2.41 | 84.4 | 90.22 | 1.57 | 90.33 | **91.12** | **0.89** | **91.19** |
| 8 | SuDA | **85.25** | 2.49 | **85.42** | 90.79 | 1.82 | **90.89** | **90.68** | **0.88** | **90.72** |
| | SuDA Adapt | 84.93 | 2.52 | 85.02 | **90.83** | 1.9 | 90.87 | 90.58 | 1.17 | 90.65 |
| 12 | SuDA | **84.68** | 2.87 | **84.9** | **87.29** | 3.07 | **87.38** | 89.48 | 1.44 | 89.52 |
| | SuDA Adapt | 84.22 | 2.97 | 84.44 | 87.1 | 3.24 | 87.19 | **89.88** | 1.69 | **89.95** |

Table 17: Performance of SuDA under the adaptive attack with different poison ratios.

| Poison Ratio | Defense | CIFAR-10 | | | FMNIST | | | SVHN | | |
|---|---|---|---|---|---|---|---|---|---|---|
| | | ACC(%) | ASR(%) | MA(%) | ACC(%) | ASR(%) | MA(%) | ACC(%) | ASR(%) | MA(%) |
| 1% | SuDA | **85.59** | 1.2 | 85.71 | **91.36** | 1.6 | **91.43** | 90.45 | 0.99 | 90.52 |
| | SuDA Adapt | 85.59 | 1.2 | 85.71 | 91.16 | 1.6 | 91.24 | **90.71** | **0.99** | **90.78** |
| 5% | SuDA | **85.2** | **2.3** | **85.31** | **91.46** | **1.44** | **91.55** | 90.72 | 1 | 90.75 |
| | SuDA Adapt | 84.28 | 2.41 | 84.4 | 90.22 | 1.57 | 90.33 | **91.12** | **0.89** | **91.19** |
| 10% | SuDA | **85.38** | 2.06 | **85.52** | 91.32 | 1.57 | 91.37 | 90.6 | **0.82** | 90.67 |
| | SuDA Adapt | 85.18 | 2.96 | 85.32 | **91.41** | **1.46** | **91.47** | **90.63** | 1.01 | **90.72** |
| 20% | SuDA | 85.53 | **2.01** | 85.71 | 90.91 | **1.64** | 90.95 | **91** | **0.76** | **91.05** |
| | SuDA Adapt | **85.78** | 2.34 | **85.85** | **91.38** | 1.71 | **91.43** | 90.55 | 0.82 | 90.58 |

Table 18: Performance of different defense algorithms under large attacker ratios.

| Defense | Atk Num = 0 | | | Atk Num = 10 | | | Atk Num = 20 | | | Atk Num = 30 | | |
|---|---|---|---|---|---|---|---|---|---|---|---|---|
| | ACC(%) | ASR(%) | MA(%) | ACC(%) | ASR(%) | MA(%) | ACC(%) | ASR(%) | MA(%) | ACC(%) | ASR(%) | MA(%) |
| FedAvg | 89.35 | **0.67** | 89.5 | 79.46 | 97.23 | 87.17 | 81.61 | 94.82 | **89.3** | 81.99 | 96.92 | **89.94** |
| RFA | 87.51 | 1.16 | 87.56 | 80.83 | 81.25 | 87.24 | 80.4 | 86.75 | 87.23 | 79.67 | 91.88 | 86.91 |
| MultiKrum | 88.33 | 1.22 | 88.4 | 88.32 | 1.35 | 88.47 | 80.35 | 88.31 | 87.3 | 81.03 | 97.01 | 88.88 |
| Coomed | 87.91 | 0.96 | 87.91 | 81.41 | 89.1 | 88.57 | 80.94 | 91.51 | 88.27 | 80.64 | 95.46 | 88.3 |
| Normclip | 79.71 | 1 | 79.65 | 73.8 | 83.57 | 79.77 | 73.16 | 87.8 | 79.43 | 73.26 | 89.8 | 79.68 |
| SuDA(ours) | **89.74** | 0.94 | **89.78** | **88.85** | **1.24** | **88.92** | **87.67** | **2.23** | 87.88 | **85.13** | **3.46** | 85.38 |

Table 19: Performance of different defense algorithms under large poison ratios.

| Defense | Poison Ratio = 20% | | | Poison Ratio = 40% | | | Poison Ratio = 60% | | | Poison Ratio = 80% | | |
|---|---|---|---|---|---|---|---|---|---|---|---|---|
| | ACC(%) | ASR(%) | MA(%) | ACC(%) | ASR(%) | MA(%) | ACC(%) | ASR(%) | MA(%) | ACC(%) | ASR(%) | MA(%) |
| FedAvg | 81.1 | 99.25 | **89.16** | 80.48 | 99.75 | **88.53** | 79.05 | 99.69 | **86.95** | 76.92 | 99.78 | 84.62 |
| RFA | 79.44 | 89.51 | 86.71 | 78.12 | 88.55 | 85.07 | 75.95 | 92.21 | 83.03 | 74.95 | 85.54 | 81.54 |
| MultiKrum | 80.36 | 99.47 | 88.37 | 79.9 | 99.75 | 87.9 | 77.88 | 99.69 | 85.67 | 74.92 | 99.78 | 82.42 |
| Coomed | 80.28 | 98.04 | 88.15 | 79.67 | 99.32 | 87.6 | 78.29 | 99.78 | 86.13 | 75.01 | 99.75 | 82.51 |
| Normclip | 71.92 | 92.34 | 78.45 | 69.27 | 93.13 | 75.62 | 66.38 | 93.55 | 72.48 | 61.66 | 93.79 | 67.31 |
| SuDA(ours) | **84.78** | **4.34** | 85.09 | **84.9** | **4.58** | 85.16 | **85.02** | **3.92** | 85.23 | **84.8** | **4.4** | **85.09** |

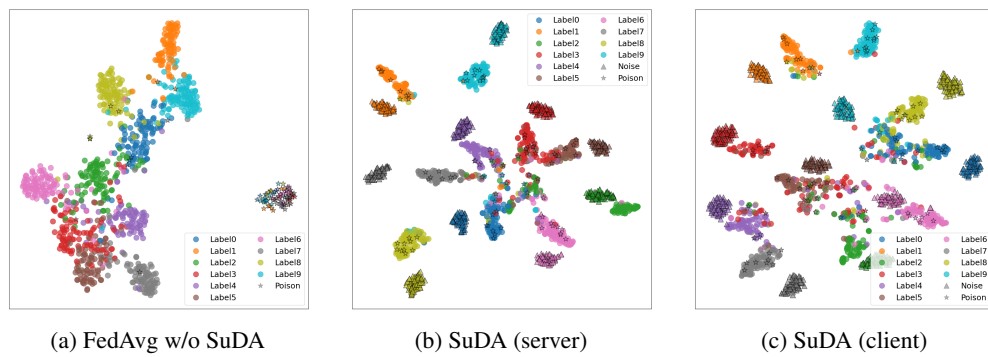

    (a) FedAvg w/o SuDA          (b) SuDA (server)          (c) SuDA (client)

Figure 8: The visualization of the feature distribution of FedAvg with (without) SuDA, at the 199-th communication round. The dots represent real data, the triangles represent noise data, the stars represent backdoored data, and different colors indicate different classes.

## F    VISUALIZATION OF FEATURE DISTRIBUTION

We exploit t-SNE (Van der Maaten & Hinton, 2008) to visualize the feature distribution, further illustrating how SuDA utilizes the noise dataset to help servers defend against backdoor attacks. Specifically, we demonstrate the feature distributions of FedAvg with (without) SuDA on the test data for 199 rounds, showcasing their respective generalization capabilities. Figure 8a shows the feature distribution of FedAvg at round 199. It can be observed that FedAvg brings the features from the same class closer together, thereby enabling the classification of different class samples. Meanwhile, the features of samples implanted with triggers are also be clustered, causing the model to misclassify them as the target class. Figures 8b and 8c represent the feature distributions of SuDA for 199 rounds. The poisoned samples are also correctly clustered together with samples of the same class.

