# OpenReview forum: "Strengthening Federated Learning: Surrogate Data-Guided Aggregation for Robust Backdoor Defense"
_ICLR.cc/2025/Conference — Submitted to ICLR 2025_

### Official Review · Reviewer_ZmhU · 2024-10-25

**Soundness:** 2
**Presentation:** 3
**Contribution:** 3
**Rating:** 5
**Confidence:** 3

**Summary:**

This paper introduces a method called “surrogate data-guided aggregation” for learning federated learning models in the presence of attackers that are trying to backdoor the model. The main idea of SuDA is to provide a surrogate dataset that is held by the central server. However, unlike most surrogate datasets, this one has nothing to do with the target (private) client data distributions---it is drawn from a noise distribution that is independent of the data. Then, the clients train their models locally on an objective that optimizes performance both on the true training data and the surrogate data. To prevent this from affecting model quality, the local clients align their representations on real data to the representations on noise. Then, the central server attempts to filter out triggered (adversarial) models that have been backdoored by reconstructing a trigger for each target class. The authors demonstrate that SuDA outperforms several defenses against trigger-based backdoor attacks.

**Strengths:**

Overall I think this paper is interesting. I liked a few things about it:

-	The problem is important (and very heavily studied)
-	The approach is creative, I liked the use of a completely noisy surrogate dataset
-	The representation alignment idea was also nice to make training on noisy data not harm model utility too much

**Weaknesses:**

I want to clarify that I generally liked the paper. I do have some concerns that I think would need to be addressed for an ICLR publication though.

1) The theoretical contributions didn't seem very strong. Can you state Theorem 4.3 more precisely? What do you mean by “elicits the bounded statistical robustness”? Do you mean there exists a lower bound on statistical robustness? Is the lower bound greater than zero?

Moreover, I didn’t understand why I should care about the theoretical result, in the sense that I would expect statistical robustness to have a nonzero lower bound for almost any reasonable classifier and dataset (since it’s an expectation over the distribution), regardless of whether it is trained with SuDA or not. Can you explain whether this theorem is providing unexpected insights? For instance, if you train a classifier without special defenses, and there are adversarial clients trying to backdoor the model, does the resulting model typically have zero statistical robustness?

2) The baselines in the evaluation seem a bit outdated. The most recent one is FLAME from 2022. I would have liked to see some more recent competitive baselines included, such as:
-	S. Wang et al, “Towards a defense against federated backdoor attacks…” (2023)
-	K. Kumari et al, “BayBFed: …” (2023)
-	T. Krauss et al, “MESAS: …” (2023)
The paper also seems to be missing a discussion of more recent FL defenses (and attacks). I’m a little worried that the evaluation and related work aren't deep enough for a FL backdoor paper.

3) If you only need random noise for the surrogate data, why do you pass it through a GAN instead of just drawing from a Gaussian and using that as the surrogate dataset? Also why do you instantiate this data as pure noise rather than some public dataset from a somewhat-related domain? Your evaluation examines part of this question, in the sense of using pure noise and not participating in training, but it would be better to also evaluate the effect of using pure noise (not passing it through a GAN), but still participating in training. (As well as possibly using public data for the surrogate training dataset). Of course I understand this won’t always be possible for all FL problems.

4) Smaller issues:

* Just to clarify---I don’t think the paper explicitly says this, but if a model is filtered out (because a small trigger is found), that model is not included in the subsequent aggregation, right? Can you please explicitly explain in the paper what happens after model filtering?

* The notation was a bit confusing. Y_t is listed as the target label, but t is also used as the round in FL.

**Questions:**

1) Could you clarify the significance of the theoretical result?

2) Can you provide results from more recent defense baselines? (and update your discussion of related work accordingly, there are a lot more papers that I didn't mention)

3) Can you explain why you construct the surrogate data by passing noise through an untrained GAN? Is there a simpler way of doing this?

---

> ### Author Response · Authors · 2024-12-03
>
> > **Q1:** Could you clarify the significance of the theoretical result?
>
> ***Ans for Q1):*** The defined statistical robustness refers to the expected distance from each sample to the closest adversarial example. Therefore, it is always greater than 0. For robust classifiers, we want the lower bound of their statistical robustness to be as large as possible. The theoretical result proves the lower bound of models trained according to the proposed objective. For classifiers without special defenses, backdoor attacks allow adversarial examples to be found for any sample with only minor modifications, resulting in very low statistical robustness.
>
> > **Q2:** Can you provide results from more recent defense baselines?
>
> ***Ans for Q2):*** Thanks for your comments. Due to time constraints, we will discuss and compare with more SOTA methods in the revised paper.
>
> > **Q3:** Can you explain why you construct the surrogate data by passing noise through an untrained GAN? Is there a simpler way of doing this?
>
> ***Ans for Q3):*** In Appendix E.4, we employ two additional generated datasets to replace the original surrogate dataset produced by StyleGAN, including one generated by a simple CNN and another generated by upsampling pure Gaussian noise. The experimental results indicate that all three methods can provide powerful defense capabilities to the server, with the surrogate data generated by Style-GAN performing slightly better in most cases.
>
> > **Q4:** Can you please explicitly explain in the paper what happens after model filtering?
>
> ***Ans for Q4):*** After model filtering, malicious models do not participate in subsequent aggregation. Following your valuable comments, we have added the above explaination to our revision.

---

### Official Review · Reviewer_1Ytg · 2024-11-03

**Soundness:** 3
**Presentation:** 3
**Contribution:** 3
**Rating:** 5
**Confidence:** 5

**Summary:**

This paper introduces SuDA, a defense mechanism against backdoor attacks in Federated Learning. SuDA employs a surrogate dataset composed of pure noise to independently evaluate local models, effectively mitigating the risk of traditional defense metrics being manipulated by malicious clients. By training local models on both real and surrogate data and aligning their feature distributions, SuDA reconstructs potential triggers and identifies backdoored models before aggregation. Experiments conducted on three image datasets (CIFAR-10, FMNIST, and SVHN) across various attack scenarios demonstrate SuDA's effectiveness, particularly in settings with a high proportion of malicious clients.

**Strengths:**

- SuDA presents a new approach by using surrogate noise data to independently assess local models, reducing the impact of malicious models on defense metrics.
- The paper provides a theoretical analysis connecting the model's generalization performance with distribution shift, justifying the use of feature distribution alignment between real and surrogate data.
- Comprehensive experiments explore different hyperparameters and attack scenarios, validating SuDA’s efficacy across diverse conditions.

**Weaknesses:**

- The evaluation only includes fixed-trigger attacks, limiting insight into SuDA's robustness against more advanced adaptive attacks like 3DFed[1], A3FL[2], and IBA[3] (optimized-trigger attacks). How might SuDA handle these more sophisticated attacks?
- Several surrogate generation methods are examined, but potential limitations of using pure noise data as a surrogate are not discussed. Could a more representative surrogate dataset (while maintaining privacy) enhance defense performance in certain cases?
- The comparison lacks state-of-the-art defenses such as DeepSight[4], FLDetector[5], FLIP[6], and BackdoorIndicator[7], making it difficult to evaluate SuDA’s relative performance and computational efficiency.

**Questions:**

- Have you explored other distance metrics beyond the $L_1$ norm for measuring trigger size? How might different metrics (e.g., $L_0$, $L_2$, or $L_{\infty}$ norms) impact SuDA’s effectiveness?
- What is SuDA's sensitivity to the hyperparameter $\lambda = 0$ in Table 15? Were any automated methods for tuning this parameter considered?
- The paper mentions that SuDA can detect backdoor models even with a small number of malicious clients. Could you elaborate on the minimum number required for effective detection?
- How does SuDA address non-iid data distributions among clients? Are there any assumptions or strategies for handling this challenge within the defense framework?
- The experimental setup diverges from previous studies in backdoor attack/defense research (DBA, FLIP[6], A3FL[2], and IBA[3]) which use 100 clients and 4 attackers with non-iid data ($\alpha = 0.5$). How would SuDA perform under these standard conditions?
- What is the value of $n_k$ in the aggregation phase when clients use surrogate data?

**References:**

[1]. Li, Haoyang, et al. "3DFed: Adaptive and extensible framework for covert backdoor attack in federated learning." 2023 IEEE Symposium on Security and Privacy (SP). IEEE, 2023.

[2]. Zhang, Hangfan, et al. "A3FL: Adversarially adaptive backdoor attacks to federated learning." Advances in Neural Information Processing Systems 36 (2024).

[3]. Nguyen, Thuy Dung, et al. "IBA: Towards irreversible backdoor attacks in federated learning." Advances in Neural Information Processing Systems 36 (2024).

[4]. Rieger, Phillip, et al. "Deepsight: Mitigating backdoor attacks in federated learning through deep model inspection." arXiv preprint arXiv:2201.00763 (2022).

[5]. Zhang, Zaixi, et al. "Fldetector: Defending federated learning against model poisoning attacks via detecting malicious clients." Proceedings of the 28th ACM SIGKDD Conference on Knowledge Discovery and Data Mining. 2022.

[6]. Zhang, Kaiyuan, et al. "FLIP: A provable defense framework for backdoor mitigation in federated learning." International Conference on Learning Representations (2023).

[7]. Li, Songze, and Yanbo Dai. "BackdoorIndicator: Leveraging OOD Data for Proactive Backdoor Detection in Federated Learning." arXiv preprint arXiv:2405.20862 (2024).

---

> ### Author Response · Authors · 2024-12-03
>
> > **Q1:** What is SuDA's sensitivity to the hyperparameter $\lambda=0$ in Table 15? Were any automated methods for tuning this parameter considered?
>
> ***Ans for Q1):*** The results in Table 15 demonstrate that SuDA is not sensitive to the align weight $\lambda$, and it can achieve good performance within a wide range of $\lambda$. In our experiments, the parameter $\lambda$ is set to a default value of 1.
>
> > **Q2:** How does SuDA address non-iid data distributions among clients? Are there any assumptions or strategies for handling this challenge within the defense framework?
>
> ***Ans for Q2):*** SuDA introduces surrogate data and aligns feature distributions. On one hand, it increases the amount of training data and reduces the imbalance between labels in the training data. On the other hand, aligning the feature distributions reduces client drift. Therefore, SuDA performs well in the Non-IID setting.
>
> > **Q3:** What is the value of $n_k$ in the aggregation phase when clients use surrogate data?
>
> ***Ans for Q3):*** The size of the surrogate dataset in our experiments is 2000. We also conduct ablations on the sensitivity of the size of the surrogate dataset and report results in Table 10.
>
> > **Q4:** Explore other distance metrics and experimental settings.
>
> ***Ans for Q4):*** Thanks for your comments. Due to time constraints, we will discuss and compare with more settings in the revised paper.

---

### Official Review · Reviewer_gY68 · 2024-11-04

**Soundness:** 2
**Presentation:** 2
**Contribution:** 2
**Rating:** 5
**Confidence:** 4

**Summary:**

The paper introduces a new Federated Learning (FL) backdoor defense. Traditional defense mechanisms often rely on empirical metrics that can be skewed by malicious client contributions, leading to ineffective defenses. The proposed Surrogate Data-Guided Aggregation for Robust Backdoor Defense (SuDA) addresses this by using surrogate data shared between the server and clients. SuDA employs feature alignment techniques to minimize distributional shifts between local and surrogate data, aiming to improve model generalization and robustness against backdoor attacks. Empirical results show that SuDA is effective on standard datasets (CIFAR-10, FMNIST, SVHN) and in adaptive attack scenarios.

**Strengths:**

* The surrogate data-based approach is flexible and could be adapted to different FL architectures (e.g., centralized or decentralized) as long as clients and servers can communicate surrogate data.
* The proposed defense mechanism is resilient to different types of backdoor patterns, and its design makes it less dependent on assumptions about specific attack triggers.

**Weaknesses:**

* Training with surrogate noise data could lead to over-regularization or reduced performance, especially in cases where the model requires specialized training on specific, non-noise-related features.
* The inclusion of surrogate data and trigger reconstruction introduces additional computational and communication overhead, which may limit scalability, particularly in resource-constrained FL setups.
* SuDA assumes that backdoor patterns can be identified and reconstructed using the surrogate data approach. However, adaptive attackers could vary the backdoor trigger dynamically, evading this detection method.
* The defense’s effectiveness relies on hyperparameters such as the alignment parameter for feature distribution. Tuning may be complex and environment-dependent.
* Using surrogate noise data in model training could potentially degrade the performance of benign models, particularly when training on highly specialized or sensitive datasets.

**Questions:**

* Could SuDA be adapted to settings with non-image data, like natural language or time-series data?
* How well does SuDA handle backdoor attacks that dynamically change over time rather than using a fixed trigger pattern?

---

> ### Author Response · Authors · 2024-12-03
>
> > **Q1:** The inclusion of surrogate data and trigger reconstruction introduces additional computational and communication overhead, which may limit scalability, particularly in resource-constrained FL setups.
>
> ***Ans for Q1):*** In Appendix E.9, we discuss the additional overhead introduced by the proposed method and further present a more efficient SuDA-Efficient, making the additional overhead acceptable.
>
> > **Q2:** How well does SuDA handle backdoor attacks that dynamically change over time rather than using a fixed trigger pattern?
>
> ***Ans for Q2):*** In the proposed SuDA, triggers are independently reconstructed in each round based on the submitted local models. As long as the model has been affected by a backdoor attack, its trigger can be reconstructed, regardless of whether the trigger dynamically changes during the process.
>
> > **Q3:** The defense’s effectiveness relies on hyperparameters such as the alignment parameter for feature distribution. Tuning may be complex and environment-dependent.
>
> ***Ans for Q3):*** The results in Table 15 demonstrate that SuDA is not sensitive to the align weight $\lambda$, and it can achieve good performance within a wide range of $\lambda$. In our experiments, the parameter $\lambda$ is set to a default value of 1.
>
> > **Q4:** Using surrogate noise data in model training could potentially degrade the performance of benign models.
>
> ***Ans for Q4):*** Directly introducing surrogate data that differs from the true data distribution during training can indeed potentially degrade the performance of benign models. As detailed in Section 4.3, SuDA addresses this issue by aligning feature distributions, and subsequent experiments validate the effectiveness of this method.
>
> > **Q5:** Could SuDA be adapted to settings with non-image data, like natural language or time-series data?
>
> ***Ans for Q5):*** Adapting SuDA to settings with non-image data is an interesting problem, we leave it for future work due to the time limitation.

---

### Official Review · Reviewer_8Rwn · 2024-11-06

**Soundness:** 2
**Presentation:** 2
**Contribution:** 2
**Rating:** 3
**Confidence:** 4

**Summary:**

The paper aims to design a defense method against backdoor attack under the federated learning setting. Without calculating predefined metrics related to local models and modifying the server’s aggregation rule accordingly, the paper proposes SuDA to evaluate the local models based on the surrogate data, which is constructed by pure noise. Besides, SuDA aligns local data with surrogate data in the representation space to ensure the generalizability of local models. Several experiments are conducted to demonstrate the performance of SuDA.

**Strengths:**

The paper propose a backdoor defense method that is independent of local models.
Theoretical analysis are provided for the generalizability of local models.

**Weaknesses:**

* The referenced methods are outdated - most of the referenced papers are before 2022. Plenty of recent studies focus on FL backdoor defense ([1-4] listed below are just a small subset). The authors should discuss and compare with more SOTA methods.
* The overview of SuDA in 4.1 referred equations that are not introduced previously, which may cause confusion.
* Observation 4.1 is provided without any empirical results validating it.
* Eqn 3 cannot cover all types of triggers. For example, it cannot represent the trigger that only change a small region in bottom right corner, as all pixels in x_posion will be changed

[1] Zhu, Chengcheng, et al. "ADFL: Defending backdoor attacks in federated learning via adversarial distillation." Computers & Security 132 (2023): 103366.
[2] Kumari, Kavita, et al. "Baybfed: Bayesian backdoor defense for federated learning." 2023 IEEE Symposium on Security and Privacy (SP). IEEE, 2023.
[3] Wang, Shuaiqi, et al. "Towards a defense against federated backdoor attacks under continuous training." arXiv preprint arXiv:2205.11736 (2022).
[4] Nguyen, Thuy Dung, et al. "Iba: Towards irreversible backdoor attacks in federated learning." Advances in Neural Information Processing Systems 36 (2024).

**Questions:**

* Why the surrogate dataset is generated by Style-GAN rather than a more simpler model. Is there a specific reason?
* Why the surrogate data share the same labels with training data? Can they have total different label set?
* What is the definition of \alpha in 5.1?

---

> ### Author Response · Authors · 2024-12-03
>
> > **Q1:** The authors should discuss and compare with more SOTA methods.
>
> ***Ans for Q1):*** Thanks for your comments. Due to time constraints, we will discuss and compare with more SOTA methods in the revised paper.
>
> > **Q2:** The overview of SuDA in 4.1 referred equations that are not introduced previously, which may cause confusion.
>
> ***Ans for Q2):*** In Section 4.1, we introduce the SuDA framework to provide readers with an overall understanding of the proposed method before delving into the details. The equations cited in Section 4.1 are also explained in detail in the subsequent sections.
>
> > **Q3:** Observation 4.1 is provided without any empirical results validating it.
>
> ***Ans for Q3):*** In Figure 2, we present potential triggers for the target label (second column) and non-target labels (third column) reconstructed from poisoned data. We can observe that the potential triggers corresponding to the target label are significantly fewer than those corresponding to non-target labels, which validates our observation.
>
> > **Q4:** Eqn 3 cannot cover all types of triggers. For example, it cannot represent the trigger that only changes a small region in bottom right corner, as all pixels in x_posion will be changed.
>
> ***Ans for Q4):*** Equation 3 can represent the proposed trigger type. It only requires setting the small region corresponding to the trigger in $M$ to 1, while setting the rest of the area to 0.
>
> > **Q5:** Why the surrogate dataset is generated by Style-GAN rather than a more simpler model. Is there a specific reason?
>
> ***Ans for Q5):*** In Appendix E.4, we employ two additional generated datasets to replace the original surrogate dataset produced by StyleGAN, including one generated by a simple CNN and another generated by upsampling pure Gaussian noise. The experimental results indicate that all three methods can provide powerful defense capabilities to the server, with the surrogate data generated by Style-GAN performing slightly better in most cases.
>
> > **Q6:** Why the surrogate data share the same labels with training data? Can they have total different label set?
>
> ***Ans for Q6):*** Surrogate data share the same labels as the training data because we want the surrogate data to represent the real data and align their feature distributions. If their label sets are different, the feature distributions cannot be effectively aligned.
>
> > **Q7:** What is the definition of \alpha in 5.1?
>
> ***Ans for Q7):*** $\alpha$ is the parameter of the commonly used Non-IID partition method Latent Dirichlet Sampling.

---

### Meta-Review · Area_Chair_YzAQ · 2024-12-20

**Metareview:**

This paper received four negative ratings, with all reviewers generally inclined to reject it. The paper proposes a defense method against backdoor attacks in the federated learning setting. However, it relies on outdated methods, with most of the cited studies predating 2022, and lacks comparison to recent state-of-the-art (SOTA) defenses such as DeepSight and FLDetector. Theoretical contributions are unclear, and the concept of "bounded statistical robustness" requires further explanation. The use of surrogate noise data may lead to over-regularization, potentially degrading performance, and the additional computational overhead limits scalability, especially in resource-constrained environments. Furthermore, SuDA’s assumption of static backdoor patterns makes it vulnerable to adaptive attacks, which the evaluation fails to address. The paper also overlooks the potential benefits of using more representative surrogate datasets or public data. Overall, while the paper shows some positive benefit, it requires significant revisions for ICLR publication, and none of the reviewers have raised their scores after reviewing the authors' response. Therefore, the Area Chair (AC) recommends rejecting the paper.

**Additional Comments On Reviewer Discussion:**

This paper received four negative ratings, with all reviewers generally inclined to reject it. While the paper shows some positive benefit, it requires significant revisions for ICLR publication, and none of the reviewers have raised their scores after reviewing the authors' response.

---

### Decision · Program_Chairs · 2025-01-22

Reject